# Neural Latent Arbitrary Lagrangian-Eulerian Grids for Fluid-Solid Interaction

**Shilong Tao**[1], **Zhe Feng**[1], **Shaohan Chen**[1], **Weichen Zhang**[2]
**Zhanxing Zhu**[3,*], **Yunhuai Liu**[1,*]
[1]School of Computer Science, Peking University
[2]Global Innovation Exchange, Tsinghua University
[3]School of Electrical and Computer Science, University of Southampton

## Abstract

Fluid-solid interaction (FSI) problems are fundamental in many scientific and engineering applications, yet effectively capturing the highly nonlinear two-way interactions remains a significant challenge. Most existing deep learning methods are limited to simplified one-way FSI scenarios, often assuming rigid and static solid to reduce complexity. Even in two-way setups, prevailing approaches struggle to capture dynamic, heterogeneous interactions due to the lack of cross-domain awareness. In this paper, we introduce **Fisale**, a data-driven framework for handling complex two-way **FSI** problems. It is inspired by classical numerical methods, namely the Arbitrary Lagrangian–Eulerian (**ALE**) method and the partitioned coupling algorithm. Fisale explicitly models the coupling interface as a distinct component and leverages multiscale latent ALE grids to provide unified, geometry-aware embeddings across domains. A partitioned coupling module (PCM) further decomposes the problem into structured substeps, enabling progressive modeling of nonlinear interdependencies. Compared to existing models, Fisale introduces a more flexible framework that iteratively handles complex dynamics of solid, fluid and their coupling interface on a unified representation, and enables scalable learning of complex two-way FSI behaviors. Experimentally, Fisale excels in three reality-related challenging FSI scenarios, covering 2D, 3D and various tasks. The code is available at `https://github.com/therontau0054/Fisale`.

## 1 Introduction

Fluid-Solid Interaction (FSI) refers to a complex coupled phenomenon in which solids undergo motion or deformation under the action of surrounding flowing fluid, and reciprocally alter the pressure and velocity distribution within the fluid (Hou et al., 2012). This kind of problems are ubiquitous in real-world scenarios, spanning a wide range of practical applications. From biomedical engineering, such as blood flow interacting with vessel valves (Bazigou & Makinen, 2013; Enderle & Bronzino, 2012), to aerospace and civil engineering involving structural responses to fluid forces (Prasad & Wanhill, 2017; Zhang, 2011), FSI plays a pivotal role in both analysis and design.

FSI problems encompass the interplay between fluid flow, solid deformation, and their intricate coupling dynamics. These interactions are typically governed by a tightly coupled system of partial differential equations (PDEs) (Belytschko, 1980). Accurately and efficiently solving them is crucial for practical applications (Kopriva, 2009; Roubíček, 2013). However, due to nonlinear materials, moving interfaces, and strong coupling relations, analytic solutions to these PDEs are typically intractable (Xing, 2019; Génevaux et al., 2003). Thus, these FSI problems are generally discretized into meshes and solved using numerical methods such as the Immersed Boundary Method (IBM) (Peskin, 2002) and Arbitrary Lagrangian-Eulerian (ALE) (Hirt et al., 1974) method. These approaches handle the coupled system either monolithically (Heil et al., 2008), treating fluid and solid as a unified domain, or through partitioned iteration (Degroote et al., 2009), where each subdomain is solved separately with interface data exchanged iteratively. While effective in many cases, both approaches

---

*Correspondence: yunhuai.liu@pku.edu.cn, z.zhu@soton.ac.uk

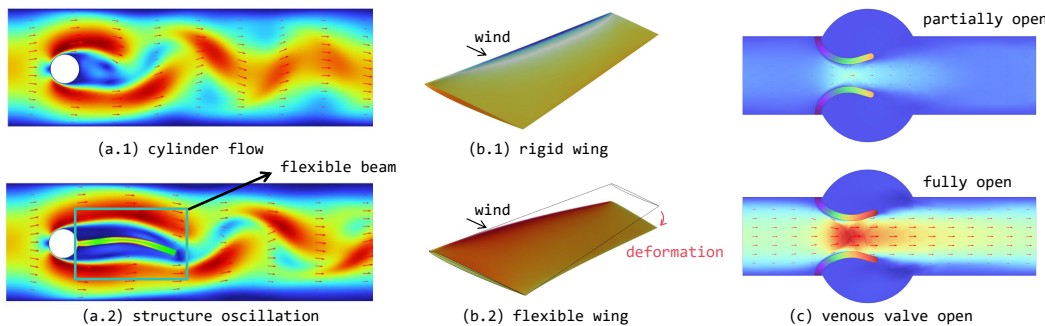

Figure 1: Fluid-solid interaction scenarios. (a.1) and (a.2) depict flow-around-body scenarios; (b.1) and (b.2) focus on aerodynamic analysis of wings; (c) illustrates the periodic dynamics of a venous valve; (a.1) and (b.1) represent one-way FSI cases, while the others involve two-way FSI.

are computationally expensive (Umetani & Bickel, 2018), strongly mesh-dependent (Berzins, 1999; Burkhart et al., 2013), and facing stability issues (Grétarsson et al., 2011; Paidoussis, 2005).

Recently, deep learning has emerged as a powerful tool for solving PDEs (Li et al., 2021; Lu et al., 2021), thanks to its strong capacity to effectively capture nonlinear input-output mappings. Once trained, it can offer significantly faster inference than traditional solvers (Han et al., 2018; Tao et al., 2025a;b; Tang et al., 2026). However, in the context of FSI, most existing models primarily focus on one-way FSI scenarios, where the solid is typically treated as static and rigid. For example, many studies have explored airfoil design tasks (Bonnet et al., 2022; Valencia et al., 2025). As shown in Figure 1(b.1), the aircraft wing is typically modeled as a static rigid body. This allows the solver to treat the wing region as a fixed, undeformable inner boundary and focus solely on the fluid domain, significantly reducing the complexity. However, as shown in Figure 1(b.2), real wings are often made of flexible materials and may undergo noticeable deformation under aerodynamic loads (Aono et al., 2010; Eppler, 2012). In such cases, the fluid-solid interface becomes dynamic, and the coupling relation grows more complex. Similar assumptions also appear in other benchmarks like cylinder flow (Pfaff et al., 2020; Li et al., 2025) and car design tasks (Elrefaie et al., 2024a;b), where structural flexibility is ignored. Thus, *how to effectively handle the evolution of fluid and solid, and capture their dynamic coupling interactions* remain unanswered when learning two-way FSI problems.

Regarding existing two-way cases, GNN-based models have simulated several simple rigid motion scenarios (Sanchez-Gonzalez et al., 2020; Li et al., 2019). While message passing underpins GNNs, it is inherently stateless and undifferentiated, making it difficult to distinguish inter- and intra-domain information in complex deformation scenarios (Hou et al., 2019). Moreover, its local reception also falls short in global modeling (Li et al., 2023d). A closely related work to ours is CoDA-NO (Rahman et al., 2024), a multiphysics neural operator that tackles a classic two-way FSI problem, structure oscillation (Figure 1(a.2)) (Turek & Hron, 2006), by partitioning the input domain along physical variable channels and learning the global mapping through codomain-wise attention. However, this variable-wise strategy maintains a monolithic view and lacks explicit handling of the dynamic fluid–solid interface caused by structural deformation. More broadly, most neural operators struggle to simultaneously learn the distinct behaviors and bidirectional dependencies of fluid and solid domains under such monolithic modeling approaches. As a result, two-way FSI still remains underexplored.

To effectively capture the evolution of solid and fluid states and their complex interactions, we propose Fisale, a purely data-driven framework inspired by classical numerical methods: ALE and partitioned coupling algorithm. ALE provides a unified representation for cross domains via mesh motion, while partitioned strategies decouple fluid and solid domains for separate processing and reduced nonlinearity. In our design, recognizing the importance of the coupling interface, we explicitly model it as a separate component on par with the solid and fluid. This enables the model to better capture the coupled dynamics. We then introduce multiscale latent ALE grids, onto which fluid, solid, and interface states are interpolated. These grids serve as unified multi-physics embeddings that encode spatial and physical quantities of the FSI system. Eventually, instead of monolithic updates, we introduce a Partitioned Coupling Module (PCM) that mirrors the logic of classical partitioned solvers. By breaking the nonlinear problem into sequential substeps, it progressively captures evolutions and complex interdependencies through deep iteration. Our contributions are summarized as follows:

- We propose to solve two-way FSI problems by separately handling the evolution of different physical domains. In particular, we treat the coupling interface as an individual component, on par with fluid and solid, enabling a more effective capture of cross-domain interactions.

- We propose Fisale, a purely data-driven framework inspired by ALE and partitioned coupling methods. It explicitly models fluid, solid, and their interface dynamics through multiscale latent ALE grids and stacked Partitioned Coupling Modules (PCM).

- Fisale achieves consistent state-of-the-art across three reality-related challenging FSI tasks, particularly in scenarios involving large deformation and complex interaction.

**Related Work** Deep learning has recently gained traction in addressing scientific problems across various fields, including initial explorations into FSI problems. A common strategy is to hybridize traditional solvers with neural networks. For instance, in partitioned frameworks, deep learning models may replace either the fluid or solid solver (Xiao et al., 2024a; Zhu et al., 2019; Mazhar et al., 2023; Xu et al., 2024; Liu et al., 2024), while the other remains conventional. Alternatively, neural networks are integrated into solvers to accelerate costly steps like velocity estimation (Fan & Wang, 2024), interface force prediction (Zhang et al., 2022; Li et al., 2023a) or control parameter approximation (Takahashi et al., 2021). Different from hybrid, purely data-driven models bypass traditional solvers entirely and learn physical dynamics directly from data, sometimes with the guidance of physical priors. One prominent class is deep reduced-order models (ROMs) (Han et al., 2022; Ashwin et al., 2022), which are built on the assumption that FSI dynamics evolve on a low-dimensional manifold (Lee et al., 2024). These methods use autoencoders (Zhai et al., 2018), PCA (Maćkiewicz & Ratajczak, 1993), or POD (Berkooz et al., 1993) to reduce dimensionality, and utilize neural networks to model the evolution in low-dimensional space efficiently. Another widely studied direction is Physics-Informed Neural Networks (PINNs) (Raissi et al., 2019), which embed the governing equations of the FSI problems directly into the loss function (Wang et al., 2021; Cheng et al., 2021; Chenaud et al., 2024), enabling the model to learn solutions that satisfy physical laws and serve as an instance-specific solver after trained. To address generalization across geometries and conditions, neural operators (Boullé & Townsend, 2024) learn mappings between function spaces, offering mesh-independent PDE solvers (Wu et al., 2024; Hao et al., 2023; Li et al., 2023d; 2025). These methods have also been extended to multi-physics problems like FSI (Rahman et al., 2024) by learning codomain-wise operator along physical variable channels. Complementarily, GNN-based simulators leverage mesh (Pfaff et al., 2020; Feng et al., 2026) or particle (Sanchez-Gonzalez et al., 2020; Li et al., 2019) connectivity to learn local interactions and propagate across physical domains.

However, most of these studies remain limited to one-way FSI scenarios, where the solid is assumed rigid and static, greatly simplifying the coupling dynamics. For the relatively few studies tackling two-way FSI, current methods, such as GNN-based simulators (Pfaff et al., 2020; Sanchez-Gonzalez et al., 2020; Li et al., 2019; Feng et al., 2026) and multi-physics neural operators (Rahman et al., 2024), struggle to effectively capture coupling behavior due to the undifferentiated and monolithic modeling. As a result, effectively learning the evolution of deformable solids and surrounding fluid flows, and capturing their complex coupling relations remain an underexplored challenge.

## 2 PRELIMINARIES

**Arbitrary Lagrangian-Eulerian Method** Lagrangian and Eulerian descriptions are two primary views for physical simulations. While the former tracks the motion trajectories of material particles (Dym et al., 1973), the latter observes physical quantities at fixed spatial locations (Morrison, 2013). Accordingly, solids are typically simulated using Lagrangian meshes that follow material deformation and motion, whereas fluids are often discretized using Eulerian grids to accommodate complex flow behavior and topological changes, making their coupling both common in nature but difficult to simulate (Xie et al., 2023; Axisa & Antunes, 2006). The Arbitrary Lagrangian–Eulerian (ALE) method (Hirt et al., 1974) combines both descriptions and is widely used in FSI simulations (Takashi & Hughes, 1992; Donea et al., 1982). The core idea is defining a mesh velocity $\mathbf{v}_g$ that moves independently of the material and fixed domain, enabling flexible mesh motion and improved stability. The mesh velocity $\mathbf{v}_g$ is typically equal to the material velocity $\mathbf{v}$ in solid domains, while obtained by Laplacian smoothing (Field, 1988) in fluid regions:

$$\nabla \cdot (\gamma \nabla \mathbf{v}_g) = \mathbf{0} \tag{1}$$

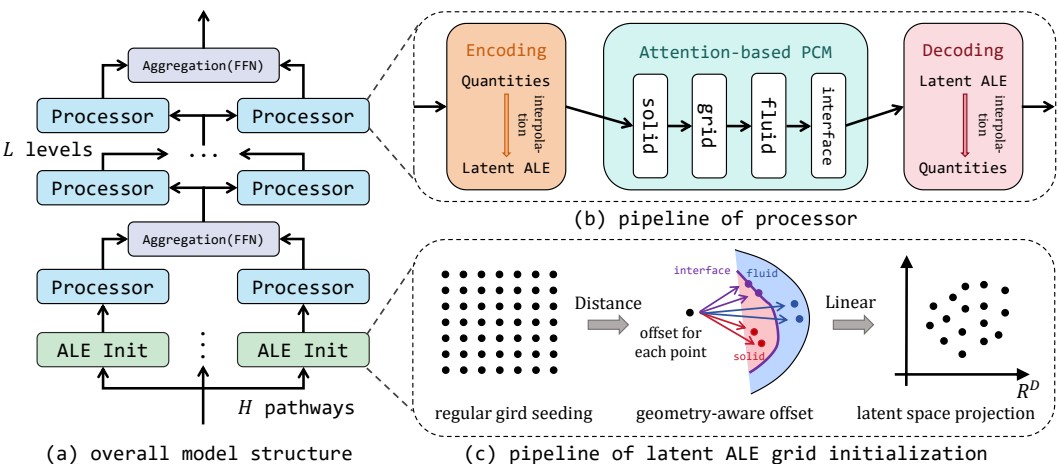

Figure 2: The overview of Fisale. (a) describes the overall structure of Fisale; (b) depicts the pipeline of the processor; (c) shows the pipeline of the latent ALE grid initialization.

where $\gamma$ is a weighting coefficient and interface velocities serve as Dirichlet boundary conditions. This flexibility of mesh motion enables ALE to accommodate unified representations for heterogeneous domains, providing a strong foundation for learning-based modeling of evolution and interaction.

**Partitioned Coupling Algorithm** The partitioned coupling algorithm is widely used in FSI simulations (Li et al., 2016; Degroote et al., 2008). By decoupling the fluid and solid fields, the coupled system is reformulated into two smaller, typically better-conditioned subproblems that can be solved using standard solvers. In contrast to monolithic approaches, where a large, fully coupled nonlinear system is solved simultaneously, the partitioned strategy reduces the dimension and complexity of the global system matrix and often yields improved numerical properties within each subdomain. A typical solution procedure (Placzek et al., 2009) consists of: (1) solving the solid subproblem based on the current solid and fluid states; (2) updating the computational mesh via ALE mesh motion method to account for the solid deformation; (3) solving the fluid subproblem on the updated mesh and solid state; and (4) matching interface, repeating above procedures or advancing to the next step.

# 3 METHOD

**Problem Setup** Given a bounded open domain $\mathcal{D} \subset \mathbb{R}^d$ and a state space $\mathcal{U} \subset \mathbb{R}^N$ representing $N$ physical quantities across fluid and solid fields (in Lagrangian, Eulerian, or hybrid form), we denote $\mathbf{u}_t(\mathbf{g}) \in \mathcal{U}$ and $\mathbf{u}_{t+\Delta t}(\mathbf{g}) \in \mathcal{U}$ as system states at two time steps $t$ and $t + \Delta t$, evaluated at a generalized spatial coordinate $\mathbf{g} \in \mathbb{R}^d$. The FSI prediction problem can be formulated as $\mathbf{u}_t(\mathbf{g}) \xrightarrow{\mathcal{F}_\theta} \hat{\mathbf{u}}_{t+\Delta t}(\mathbf{g})$, where $\mathcal{F}_\theta$ represents the learned mapping function. This basic single-step prediction can be extended to several tasks like steady-state inference, as discussed in Section 4.

**Notation** For simplicity, we omit the time subscript $t$ and denote the fluid and solid observations at current time step as $\mathbf{u}_f = [\mathbf{g}_f \in \mathbb{R}^{N_f \times d}, \mathbf{q}_f \in \mathbb{R}^{N_f \times C_f}]$ and $\mathbf{u}_s = [\mathbf{g}_s \in \mathbb{R}^{N_s \times d}, \mathbf{q}_s \in \mathbb{R}^{N_s \times C_s}]$, respectively. Here, $\mathbf{g}$ represents the generalized spatial coordinates and $\mathbf{q}$ denotes the associated physical quantities. As mentioned before, we explicitly treat the interface as a separate component and denote it as $\mathbf{u}_b = [\mathbf{g}_b \in \mathbb{R}^{N_b \times d}, \mathbf{q}_b \in \mathbb{R}^{N_b \times C_b}]$, where $C_f + C_s = C_b$ and $N_f + N_s + N_b = N$.

**Overall Framework** As shown in Figure 2a and 2b, the pipeline of Fisale is formulated as:

$$\mathcal{F}_\theta = \left( \prod_{l=1}^{L} \mathcal{F}_{\theta_{\text{Aggregate}}^{(l)}} \circ \left[ \bigoplus_{h=1}^{H} \mathcal{F}_{\theta_{\text{LatentALEToOrigin}}^{(l,h)}} \circ \mathcal{F}_{\theta_{\text{PCM}}^{(l,h)}} \circ \mathcal{F}_{\theta_{\text{OriginToLatentALE}}^{(l,h)}} \right] \right) \circ \left[ \bigoplus_{h=1}^{H} \mathcal{F}_{\theta_{\text{ALEInit}}^{(h)}} \right] \quad (2)$$

where $\circ$ denotes operator composition. The full model $\mathcal{F}_\theta$ consists of $H$ parallel latent pathways corresponding to different spatial scales. Each pathway begins with an individual ALE grid initialization $\mathcal{F}_{\theta_{\text{ALEInit}}^{(h)}}$, which provides a unified representation for different domains. Then each pathway proceeds through $L$ stacked processors. At the $l$-th level of the $h$-th pathway, the processor consists

of three sequential components: an encoding step from the original space to the latent ALE space $\mathcal{F}_{\theta_{\text{OriginToLatentALE}}^{(l,h)}}$, a processing module $\mathcal{F}_{\theta_{\text{PCM}}^{(l,h)}}$ on the latent ALE grid, and a decoding step back to the original space $\mathcal{F}_{\theta_{\text{LatentALEToOrigin}}^{(l,h)}}$. To enable cross-scale communication, an aggregation module $\mathcal{F}_{\theta_{\text{Aggregate}}^{(l)}}$ is applied after each level $l$, where features from all pathways are concatenated and passed through a Feed-Forward Network (FFN) to aggregate information and produce updated features.

## 3.1 Multiscale Latent ALE Grids

Instead of directly solving FSI in heterogeneous descriptions, we introduce a set of multiscale latent ALE grids, which provide a unified representation for multiscale and cross-domain physics.

**Latent ALE Grid Initiation** The initiation of a single latent ALE grid includes two steps: regular grid seeding and geometry-aware offset. As shown in Figure 2c, we begin by initializing a regular, axis-aligned Cartesian grid $\mathbf{a} \in \mathbb{R}^{M \times d}$ over the $d$-dimensional Euclidean space $[-3.5, 3.5]^d$ via uniform sampling. Notably, we normalize the input physical domain into a $\mathcal{N}(\mathbf{0}, \mathbf{1})$; according to the 3-sigma rule (Huber, 2018), the regular grid with interval $[-3.5, 3.5]^d$ covers more than 99.95% input mesh points. Here, the grid is flattened into a sequence of length $M$, where $M = M_1 \times M_2 \times \cdots \times M_d$ denotes the total number of grid nodes, and each $M_k$ represents the number of discretization points along the $k$-th spatial axis. This regular grid is treated as a reference grid, independent of any specific geometry. Its uniform structure enables the decoupling of spatial topology from geometric variation and facilitates geometry-aware deformation in the subsequent initialization stage (Li et al., 2023c).

To incorporate geometric awareness into the grid, we deform the regular reference grid $\mathbf{a}$ by applying an offset field that reflects the spatial distribution of solid, fluid and coupling interface. We first compute direction vectors from each grid node $\mathbf{a}_i$ to points in each domain. These vectors are fed into a linear layer and then weighted by a normalized radial basis kernel to prioritize closer points. For example, the offset contributed by the fluid geometry $\mathbf{g}_f$ is given by:

$$\Delta_f(\mathbf{a}_i) = \sum_{j=1}^{N_f} \frac{\exp(\text{Linear}(-\|\mathbf{a}_i - \mathbf{g}_{f_j}\|^2))}{\sum_{j=1}^{N_f} \exp(\text{Linear}(-\|\mathbf{a}_i - \mathbf{g}_{f_j}\|^2))} (\mathbf{g}_{f_j} - \mathbf{a}_i)$$

where the subtraction is implemented by broadcast. With this offset, the weights assigned to fluid points are optimized based on distances, leading to a geometry-aware aggregation. The total offset is obtained by summing contributions from all domains: $\Delta(\mathbf{a}) = \Delta_s(\mathbf{a}) + \Delta_f(\mathbf{a}) + \Delta_b(\mathbf{a})$. Finally, the latent grid is obtained by applying the total geometry-aware offset followed by a linear projection: $\mathbf{g}_a = \text{Linear}(\mathbf{a} + \Delta(\mathbf{a}))$. This geometry-aware offset encourages grid nodes to move closer to regions of geometric interest, such as fluid-solid interfaces, by aggregating directional influences through distance-weighted kernels. Moreover, the spatial decay of the kernel attenuates the influence of distant regions, leading to weaker and more uniform updates in non-critical regions, which helps preserve the grid's smoothness and regularity. We then build edges by performing $k$-nearest neighbor ($k$-NN) search on the latent grid $\mathbf{g}_a \in \mathbb{R}^{M \times D}$, i.e., $E = k\text{NN}(\mathbf{g}_a)$. The edge set $E$ defines a latent interaction graph over the deformed grid, which supports grid's update in subsequent modules.

**Definition of *Latent ALE Grid*** During the solving process, this grid is maintained and iteratively updated in latent space according to the learned dynamics. Crucially, it adheres to the principle of the ALE method, namely, evolving with a motion that is decoupled from both the frames of material points and spatial grids, thereby enabling an intermediate behavior between Lagrangian and Eulerian viewpoints (Hirt et al., 1974). We therefore refer to this representation as the *Latent ALE Grid*.

**Multiscale** FSI phenomena are inherently multiscale (Steinhauser, 2018). For example, a flexible wing interacting with airflow involves large-scale aerodynamic forces and small-scale local deformations. Capturing such behaviors requires the ability to model and propagate information across different resolution levels. Fortunately, it is straightforward and natural in our framework to construct grids at multiple scales, simply by varying the number of grid nodes $M$ during initialization. By operating in parallel, each grid records physical context at a different level of detail. This allows coarse grids to efficiently model global structures, while fine grids focus on local interactions, together forming a hierarchical representation well-suited for multiscale FSI problems.

**Supplementary Notation** We define $\mathbf{x}_f^{(l,h)} \in \mathbb{R}^{N_f \times D}$ to be the output feature of the $l$-th level in the $h$-th pathway. The $\mathbf{x}_f^{0,h}$ corresponds to the input embedding of the observation $\mathbf{u}_f$, i.e.,

$\mathbf{x}_f^{0,h} = \text{Linear}(\mathbf{u}_f)$. The solid and interface adhere the same manner. We denote $\mathbf{g}_a^{l,h}$ as the state of the latent ALE grid at the $l$-th level of the $h$-th pathway. Since the encoding, decoding, and PCM operations are shared across levels and pathways, we omit $l$ and $h$ in the following descriptions.

## 3.2 Physical Quantities Encoding and Decoding

Before each coupling process, we project physical quantities onto the grid to enable unification across heterogeneous regions. After coupling learning, we project them back for multiscale aggregation. Both processes are applied by weighted interpolation.

**Physical Quantities Encoding** For each domain, we compute the projection weight through an attention-like manner. The interpolation weight for fluid domain $\mathbf{w}_f \in \mathbb{R}^{M \times N_f}$ is defined as:

$$\mathbf{w}_f = \mathbf{Q}\mathbf{K}^T, \quad \text{where} \quad \mathbf{Q} = \text{Linear}(\mathbf{g}_a) \quad \text{and} \quad \mathbf{K} = \text{Linear}(\mathbf{x}_f)$$

Then the fluid projection $\mathbf{p}_f \in \mathbb{R}^{M \times D}$ is conducted by: $\mathbf{p}_f = \text{Softmax}(\mathbf{w}_f)\mathbf{x}_f$.

Solid and coupling interface domains follow the same procedure with projection weights $\mathbf{w}_s$ and $\mathbf{w}_b$, projected features $\mathbf{p}_s$ and $\mathbf{p}_b$, respectively. As a result, we extend the latent ALE grid $\mathbf{g}_a$ as $\{\mathbf{g}_a, \mathbf{p}_s, \mathbf{p}_f, \mathbf{p}_b\}$. This latent tuple encodes both the grid's geometry-aware position and the surrounding physical context, serving as the input to subsequent modules. Unlike traditional discretizations that partition physical domains rigidly, we represent fluid, solid, and interface features simultaneously at each latent node. This allows the model to naturally capture cross-domain interactions and dynamics near interfaces, forming a flexible and expressive multi-physics embedding for learning.

**Physical Quantities Decoding** Let $\{\hat{\mathbf{g}}_a, \hat{\mathbf{p}}_s, \hat{\mathbf{p}}_f, \hat{\mathbf{p}}_b\} = \text{PCM}(\{\mathbf{g}_a, \mathbf{p}_s, \mathbf{p}_f, \mathbf{p}_b\})$ denote the updated features. The decoding is similar to the encoding as: $\hat{\mathbf{x}}_f = \text{Softmax}(\mathbf{w}_f^T)\hat{\mathbf{p}}_f$.

Here, the direction of $\text{Softmax}(\cdot)$ is different from the encoding to keep the sum of the interpolation weight equal to 1. To fuse features in different scales, we concatenate the decoding feature from different pathways and adopt a Feed-Forward Network (FFN) for aggregation. Then the fused features are split and taken back to their own pathways.

## 3.3 Partitioned Coupling Module

We design a Partitioned Coupling Module (PCM) that follows the process of partitioned coupling algorithm to learn the evolution of fluid, solid and their complex coupling in an iteratively deep manner. Each PCM includes four forward steps outlined in Section 2.

**Update Solid State** We first employ the cross-attention mechanism to update the solid state. Attention mechanisms (Vaswani et al., 2017) are well-suited for modeling PDE-related physical systems due to their strong capacity to capture nonlinear dependencies, long-range interactions, and perform spatial aggregation across irregular domains (Hao et al., 2023; Wu et al., 2023; Li et al., 2023b). Recent studies have further shown that attention itself can be interpreted as an integral operator, capable of approximating complex mappings across function spaces (Cao, 2021; Wu et al., 2024; Li et al., 2025). Moreover, the flattened sequence of the latent ALE grid aligns naturally with the input format required by attention mechanisms, making attention a seamless and effective choice for updating physical states across fluid, solid, and interface domains. Given current embedding on the latent ALE grid $\{\mathbf{g}_a, \mathbf{p}_s, \mathbf{p}_f, \mathbf{p}_b\}$, the update is formulated as:

$$\mathbf{Q} = \text{Linear}(\text{Concat}(\mathbf{p}_s + \mathbf{g}_a, \mathbf{p}_b + \mathbf{g}_a)), \quad \mathbf{K}, \mathbf{V} = \text{Linear}(\text{Concat}(\mathbf{p}_s + \mathbf{g}_a, \mathbf{p}_f + \mathbf{g}_a, \mathbf{p}_b + \mathbf{g}_a))$$

$$\mathbf{p}_s', \mathbf{p}_b' = \text{Chunk}\left(\tilde{\mathbf{Q}}(\tilde{\mathbf{K}}^T\mathbf{V} \cdot D^{-1})\right), \quad \text{where} \quad \tilde{\mathbf{Q}} = \text{Softmax}(\mathbf{Q}) \text{ and } \tilde{\mathbf{K}} = \text{Softmax}(\mathbf{K})$$

The query $\mathbf{Q}$ is constructed by complete solid representation and the key $\mathbf{K}$ and value $\mathbf{V}$ include the information of the whole system. The geometry of the latent ALE grid $\mathbf{g}_a$ serves as positional embedding that provides spatial prior. We concatenate the components along the length direction and adopt the linear attention which has been approved as a kind of neural operator (Cao, 2021). Through cross-attention, each solid node selectively attends to the entire system, allowing it to update its state based on both internal structural cues and external influences from the fluid and interface. The $\text{Chunk}(\cdot)$ divides the output to recover the updated solid state $\mathbf{p}_s'$ and the interface state $\mathbf{p}_b'$.

**Update Grid Coordinate** We next adopt a velocity-based Laplacian smoothing strategy, defined in Eq.1, to update the latent ALE grid coordinates in response to solid motion and deformation while preserving grid quality. Eq.1 governs the spatial diffusion of mesh velocity $\mathbf{v}_g$ and can be interpreted as a steady-state flux balance over the grid. Upon discretization, the divergence and gradient operators naturally translate into local neighbor interactions (Han et al., 2023a): $\sum_{j \in \mathcal{N}(i)} (\mathbf{v}_{g,j} - \mathbf{v}_{g,i}) = \mathbf{0}$. The velocity at each mesh point is updated based on a weighted combination of its neighboring nodes:

$$\mathbf{v}_{g,i} \leftarrow \sum_{j \in \mathcal{N}(i)} \gamma_{ij} \mathbf{v}_{g,j} \cdot (\sum_{j \in \mathcal{N}(i)} \gamma_{ij})^{-1}$$

where $\mathcal{N}(i)$ denotes the neighbors of the $i$-th node, and $\gamma_{ij}$ is a diffusion-like weight that reflects local mesh connectivity. Coincidentally, this update scheme is naturally aligned with the local message passing (Gilmer et al., 2017) used in graph-based models. We update the grid based on the neighbor relation $\mathcal{N}(i) \subset E$, which is built in the initialization of ALE. In latent space, we treat $\gamma \in [0, 1]$ as learnable parameters, where $\sum_{j \in \mathcal{N}(i)} \gamma_{ij} = 1$, and update the mesh velocity as:

$$\mathbf{v}_{g,i} \leftarrow \sum_{j \in \mathcal{N}(i)} \gamma_{ij} \text{Linear}(\text{Concat}(\mathbf{p}'_s, \mathbf{p}_f, \mathbf{p}'_b)_j)$$

Then the latent grid coordinates are explicitly updated by $\hat{\mathbf{g}}_a \leftarrow \mathbf{g}_a + \Delta t \cdot \mathbf{v}_g$. In our stacked-module architecture, we set $\Delta t = 1$ as an abstract timestep per module. Finally, to control the grid quality, without grid distortion, we conduct a geometry smoothing over the local neighborhood $\mathbf{g}'_{a,i} \leftarrow \sum_{j \in \mathcal{N}(i)} \beta_{ij} \hat{\mathbf{g}}_{a,j}$, where $\beta \in [0, 1]$ and $\sum_{j \in \mathcal{N}(i)} \beta_{ij} = 1$.

**Update Fluid State** Similar to the update of the solid state, we update fluid state as:

$$\mathbf{Q} = \text{Linear}(\text{Concat}(\mathbf{p}_f + \mathbf{g}'_a, \mathbf{p}'_b + \mathbf{g}'_a)), \quad \mathbf{K}, \mathbf{V} = \text{Linear}(\text{Concat}(\mathbf{p}'_s + \mathbf{g}'_a, \mathbf{p}_f + \mathbf{g}'_a, \mathbf{p}'_b + \mathbf{g}'_a))$$

$$\mathbf{p}'_s, \mathbf{p}''_b = \text{Chunk}\left(\tilde{\mathbf{Q}}(\tilde{\mathbf{K}}^T \mathbf{V} \cdot D^{-1})\right)$$

**Update Interface Influence** Finally, we use a self-attention mechanism to align the information across the solid, fluid, and their coupling interface regions as:

$$\mathbf{Q}, \mathbf{K}, \mathbf{V} = \text{Linear}(\text{Concat}(\mathbf{p}'_s + \mathbf{g}'_a, \mathbf{p}'_f + \mathbf{g}'_a, \mathbf{p}''_b + \mathbf{g}'_a)), \quad \mathbf{p}''_s, \mathbf{p}''_f, \mathbf{p}'''_b = \text{Chunk}\left(\tilde{\mathbf{Q}}(\tilde{\mathbf{K}}^T \mathbf{V} \cdot D^{-1})\right)$$

The self-attention operation enables mutual interaction among the three domains, allowing the model to capture dependencies and reconcile inconsistencies across the solid–fluid interface by attending to relevant features globally. Finally, these four steps form a single PCM and we stack the modules to enhance the represent capacity of the model.

## 4 EXPERIMENTS

Table 1: Summary of experiment dataset. #Mesh records the average size of discretized meshes. #Split is organized as the number of samples in training, evaluation and test sets.

| Dataset | Task | #Dim | #Mesh | #Input | #Output | #Split |
|---|---|---|---|---|---|---|
| Structure Oscillation | Single-Step Prediction | 2D | 1317 | Solid: Geometry; Fluid: Geometry, Pressure, Velocity | Solid: Geometry; Fluid: Geometry, Pressure, Velocity | 9561 1195 1196 |
| Venous Valve | Autoregressive Simulation | 2D | 1693 | Solid: Geometry, Stress; Fluid: Geometry, Pressure, Velocity | Solid: Geometry, Stress; Fluid: Geometry, Pressure, Velocity | 720 90 90 |
| Flexible Wing | Steady-State Inference | 3D | 37441 | Solid: Geometry, Material; Fluid: Geometry, Attack Angle, Velocity | Solid: Geometry, Stress; Fluid: Geometry, Pressure, Velocity | 1036 129 131 |

We evaluate Fisale on three reality-related challenging FSI scenarios. These scenarios have different dimensions, targets, scales, and complexity. Detailed benchmark information is listed in Table 1.

**Baselines** We compare Fisale with over ten advanced learning-based solvers, including Neural Operators: GeoFNO (Li et al., 2023c), GINO (Li et al., 2023d), CoDA-NO (Rahman et al., 2024), LSM (Wu et al., 2023), LNO (Wang & Wang, 2024); Transformers: Galerkin Transformer (Cao, 2021), GNOT (Hao et al., 2023), ONO (Xiao et al., 2024b) Transolver (Wu et al., 2024); GNNs: MGN (Pfaff et al., 2020), HOOD (Grigorev et al., 2023), AMG (Li et al., 2025).

**Implementation and Metrics** Fisale and all baseline models are trained and evaluated under the same protocol. For a fair comparison, we match the parameter count of each baseline to that of Fisale. For loss functions and evaluation metrics, we use Relative L2 error for the single-step prediction and steady-state inference tasks, and Root Mean Squared Error (RMSE) for the autoregressive simulation task. For experiments involving multiple physical domains, the loss term is calculated for each physical domain and combined with equal weights. All experiments are conducted on a single NVIDIA RTX 3090 GPU (24 GB) and repeated multiple times. We provide more details in Appendix C and D.

Table 2: Performance on Structure Oscillation. Relative L2 is recorded. Second-best performance is underlined.

|  | Solid | Fluid | Interface | Mean ($\downarrow$) |
|---|---|---|---|---|
| Geo-FNO | 0.0003 | 0.0387 | 0.0074 | 0.0155 |
| GINO | 0.0021 | 0.2536 | 0.0269 | 0.0942 |
| LSM | 0.0007 | 0.1951 | 0.0068 | 0.0675 |
| CoDANO | 0.0005 | 0.0703 | 0.0075 | 0.0261 |
| LNO | 0.0006 | 0.0244 | 0.0061 | 0.0104 |
| Galerkin | 0.0012 | 0.0507 | 0.0114 | 0.0211 |
| GNOT | 0.0006 | 0.0361 | 0.0076 | 0.0148 |
| ONO | 0.0012 | 0.0732 | 0.0126 | 0.0290 |
| Transolver | 0.0004 | 0.0265 | 0.0075 | 0.0115 |
| MGN | 0.0007 | 0.0282 | 0.0112 | 0.0134 |
| HOOD | 0.0006 | 0.0277 | 0.0109 | 0.0131 |
| AMG | 0.0004 | 0.0211 | 0.0051 | 0.0089 |
| **Fisale** | 0.0003 | 0.0148 | 0.0047 | **0.0066** |

## 4.1 STRUCTURE OSCILLATION

The structure oscillation problem, also called "FLUSTRUK-A", is a famous benchmark for validating classical fluid–solid interaction (FSI) solvers (Turek & Hron, 2006). It involves a thin elastic beam immersed in an incompressible, dynamic fluid that

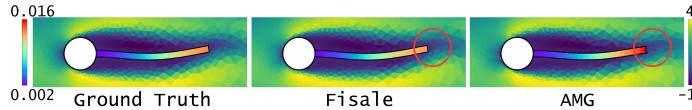

Figure 3: Local visualization of prediction results on solid displacement and fluid $x$-velocity. Red circle indicates the domain with most solid displacement and sharp fluid velocity change.

develops self-sustained, material-dependent periodic oscillations. Unlike the widely studied Cylinder-Flow benchmark (Pfaff et al., 2020) which involves no structural flexibility, FLUSTRUK-A introduces strong two-way coupling and nonlinear deformation, making it more challenging to solve accurately.

The dataset is proposed in CoDA-NO (Rahman et al., 2024) with 1000 frames and different Reynolds number $Re$. Following the convention, we train Fisale and baselines learn the mapping from current $\mathbf{u}_t$ to the next $\mathbf{u}_{t+\Delta t}$. We set $\Delta t$ as 4 frames to let the oscillation evolve adequately. The results are shown in Table 2. From the results, we can observe that Fisale achieves advanced ability, particularly in fluid domains. Compared with other baselines who model different domains in a homogeneous way, Fisale can better capture the bidirectional interactions between fluid and solid at the interface and predict the motion and shape more accurately (the tail part of the structure as shown in Figure 3). This in turn enhances the accuracy of fluid predictions. The unified latent ALE representation, combined with iterative partitioned coupling, allows Fisale to progressively resolve the nonlinear interactions between fluid and solid, resulting in more accurate and stable predictions under strongly coupled two-way FSI.

Following the convention in CoDA-NO, we explore the out-of-distribution (OOD) performance of Fisale. In previous experiments, we trained models with data that contain the Reynolds number $Re \in \{200, 400, 2000\}$. We directly test trained models on data with $Re \in \{4000\}$. As presented in Table 3, Fisale consistently performs best over strong baselines on OOD sam-

Table 3: Results of OOD test.

|  | Relative L2 ($\downarrow$) |
|---|---|
| Geo-FNO | 0.0730 |
| LNO | 0.0715 |
| GNOT | 0.0889 |
| Transolver | 0.0722 |
| MGN | 0.0742 |
| AMG | 0.0696 |
| **Fisale** | **0.0637** |

ples. This better generalization can be attributed to its latent ALE representation, which captures flow patterns across multiple spatial scales, and the partitioned coupling design, which provides a modular and robust way to update interdependent physical states. Furthermore, the explicit modeling of the coupling interface enhances the model's ability to extrapolate dynamic interface interactions under stronger nonlinear coupling, which is prevalent in high-Reynolds-number regimes.

## 4.2 VENOUS VALVE

The venous valve problem models the opening and closing dynamics of valves in veins, which are essential to maintain unidirectional blood flow in the circulatory system (Bazigou & Makinen, 2013; Enderle & Bronzino, 2012). It involves a thin, flexible leaflet interacting with a pulsatile, incompressible fluid under physiological conditions. The strong contact, large deformation, and highly transient behavior make this problem especially challenging for effective simulation (Buxton & Clarke, 2006). We learn the transient dynamics through autoregressive simulation, which are mostly explored in GNN-based works (Pfaff et al., 2020; Sanchez-Gonzalez et al., 2020). Mathematically, we simulate the trajectory as: $\hat{\mathbf{u}}_{t+1} = \mathcal{F}_\theta(\mathbf{u}_t), \hat{\mathbf{u}}_{t+2} = \mathcal{F}_\theta(\hat{\mathbf{u}}_{t+1}), \ldots, \hat{\mathbf{u}}_{t+T} = \mathcal{F}_\theta(\hat{\mathbf{u}}_{t+T-1})$. We build the venous valve simulation model based on biology-related literature and generate a dataset where each sample has different valve material properties, flow velocities, and other parameters. The variations are designed to reflect a wide range of human conditions across ages, genders, and health status. Detailed settings can be found in the Appendix B.2. The simulation time is $1s$ and $0.01s$ per interval, with 101 frames in total. Each frame records current geometry, stress, pressure, and velocity.

Table 4: Performance on Venous Valve. We record RMSE-all ($\downarrow$), the average RMSE of the whole rollout trajectory and all samples. Results of other physical quantities are listed in Table 20.

| | Solid | | Fluid | | Interface | | |
|---|---|---|---|---|---|---|---|
| | Geometry | Stress | Pressure | Velocity ($x$) | Geometry | Stress | Pressure |
| Geo-FNO | 0.3687 | 3252.51 | 124.27 | 0.1304 | 0.3948 | 5471.54 | 110.35 |
| LSM | 0.4788 | 4166.96 | 145.03 | 0.1419 | 0.4635 | 6547.88 | 122.53 |
| CoDANO | 0.6843 | 4385.24 | 171.57 | 0.1713 | 0.7806 | 6843.06 | 143.65 |
| Galerkin | 0.3471 | 3226.86 | 109.52 | 0.1025 | 0.3213 | 5093.68 | 113.34 |
| GNOT | 0.3833 | 4207.17 | 100.59 | 0.1147 | 0.3679 | 5384.59 | 128.57 |
| Transolver | 0.3262 | 3055.56 | 91.83 | 0.0901 | 0.3432 | 4941.86 | 85.18 |
| MGN | 0.5540 | 4436.56 | 166.03 | 0.1362 | 0.5391 | 6646.33 | 158.35 |
| HOOD | 0.4647 | 3616.09 | 135.67 | 0.1174 | 0.4956 | 6080.94 | 126.05 |
| AMG | 0.4029 | 3784.96 | 107.45 | 0.1199 | 0.3809 | 5432.73 | 103.43 |
| **Fisale** | **0.2794** | **2658.59** | **80.23** | **0.0768** | **0.2565** | **4365.29** | **73.31** |

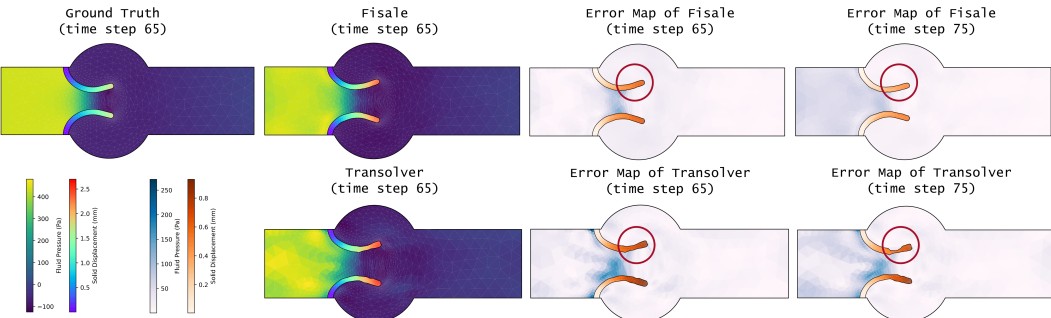

Figure 4: Visualization of ground truth and prediction results. The red circle indicates the distortion of solid shape, where Fisale can effectively handle.

As shown in Table 4, Fisale achieves the best results across all physical quantities, with particularly strong performance at the fluid–solid interface. Figure 4 visualizes predictions at two time steps. It is observable that maintaining solid shape consistency becomes increasingly challenging as the rollout progresses. Thanks to the explicit modeling of the interface and the use of a unified representation to

capture dynamic interactions, Fisale preserves solid geometry more effectively over long trajectories. This design allows it to handle cross-domain information exchange and maintain stability even in the later stages of rollout. Moreover, when fluid flows through the narrow valve opening, it generates sharp increases in pressure and velocity over short periods, making it hard to accurately predict in that region. Fisale addresses this by PCM, which decomposes this complex process into a sequence of substeps. This reduces the difficulty of modeling each physical domain and its associated quantities. In contrast, other models typically adopt a monolithic modeling strategy over the entire domain, which often struggles to capture rapid dynamic changes around the interface. This further highlights the advantage of the domain-aware design in handling complex two-way FSI phenomena.

### 4.3 FLEXIBLE WING

To better reflect real aerodynamics, we study a flexible wing scenario where the wing deforms under airflow. Unlike rigid-wing assumptions with fixed fluid–solid interfaces, flexible wings involve strong two-way coupling and nonlinear behaviors such as geometry-dependent loading and large deformations, making prediction more challenging. We treat this as a steady-state inference task: given a set of problem parameters (like the wind velocity, wing material, geometry and etc.), the model directly predicts the steady-state response. Follow the 2D rigid Airfrans (Bonnet et al., 2022), we build a 3D flexible wing dataset for evaluation. Each sample contains more than 35,000 mesh points and varies in flight and design parameters (see Appendix B.3), covering diverse flight conditions.

Table 5: Performance on Flexible Wing task. Relative L2 is recorded. Second-best performance is underlined. (a) the relative L2 error of the prediction results; (b) the visualization of prediction errors.

| | Solid | Fluid | Interface | Mean (↓) |
|---|---|---|---|---|
| Geo-FNO | 0.0207 | 0.0802 | 0.0564 | 0.0524 |
| GINO | 0.2838 | 0.5681 | 0.5715 | 0.4745 |
| CoDANO | 0.0355 | 0.1930 | 0.1002 | 0.1096 |
| LNO | 0.0173 | 0.0264 | 0.0269 | 0.0235 |
| Galerkin | 0.0396 | 0.0699 | 0.0635 | 0.0577 |
| GNOT | 0.0081 | 0.0558 | 0.0227 | 0.0289 |
| ONO | 0.1446 | 0.2362 | 0.2728 | 0.2179 |
| Transolver | 0.0051 | 0.0200 | 0.0242 | 0.0164 |
| MGN | 0.0096 | 0.0229 | 0.0281 | 0.0202 |
| HOOD | 0.0088 | 0.0218 | 0.0266 | 0.0191 |
| AMG | 0.2507 | 0.1692 | 0.2357 | 0.2185 |
| **Fisale** | **0.0042** | **0.0155** | **0.0211** | **0.0136** |

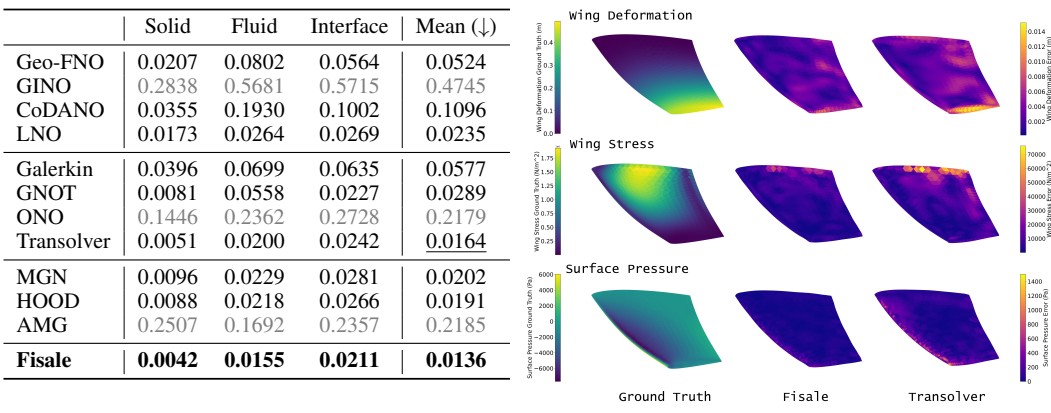

(a) The error of prediction results.          (b) The visualization of prediction errors.

As presented in Table 5, Fisale achieves the best performance across all domains. When dealing with massive mesh points, several baselines seriously degenerate due to complex dynamics. This is because dense fluid points can overwhelm solid-related information, making it difficult to capture solid changes. In turn, this also disrupts the accurate modeling of fluid evolution. The right part of Table 5 visualizes the prediction errors. For the wing structure, deformation is primarily localized at the tip, stress is concentrated at the root, and the applied wind pressure is distributed over the lateral surfaces. Each of these exhibits distinct spatial patterns driven by the interaction between the wind and the wing. Accurately capturing these patterns is therefore crucial. By modeling each domain separately, Fisale avoids letting solid information be overwhelmed by massive fluid points. Moreover, explicitly modeling the interface enables effective capture of spatially varying physics across different regions of the wing surface, thereby improving the performance in complex, large-scale scenarios.

## 5 CONCLUSION

We propose Fisale, a data-driven framework for solving complex two-way fluid-solid interaction (FSI) problems. By explicitly modeling the solid, fluid and coupling interface as separate components, and leveraging multiscale latent arbitrary Lagrangian-Eulerian (ALE) grids along with partitioned coupling modules (PCM), Fisale effectively captures nonlinear, cross-domain dynamics. Experiments on challenging tasks demonstrate the effectiveness of Fisale in solving complex two-way FSI problems.

## ETHICS STATEMENT

Our work only focuses on solving two-way fluid-solid interaction (FSI) problems with deep learning method, so there is no potential ethical risk.

## ACKNOWLEDGMENTS

This work is supported partly by the National Natural Science Foundation of China (NSFC) 62576013, National Key Research Plan under grant No.2024YFC2607404, the Jiangsu Provincial Key Research and Development Program under Grant BE2022065-1, BE2022065-3, and the Ningxia Domain-Specific Large Model Health Industry R&D No.2024JBGS001.

## REPRODUCIBILITY STATEMENT

In the main text, we provide rigorous mathematical formulations of the model architecture. The overall pipeline is described in Appendix A. Hyperparameter settings and implementation details are provided in Appendix D, and the dataset is described in Appendix B. Our code and dataset are publicly available in the GitHub repository linked in the abstract.

## LLM USAGE CLARIFICATION

We declare that the Large Language Models (LLMs) are only used for language polishing and grammar correction during the writing of this manuscript. All research content, conclusions, and data are solely produced by the authors, without subjective arguments, conclusions, or data generated by LLMs. We take full responsibility for the entire content of this paper.

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

# Contents

## A    OVERALL PIPELINE

Our overall framework is formulated as Eq.2, and we have described our design in Section 3 including multiscale latent ALE grid, physical quantities encoding, decoding and aggregation, and partitioned coupling module. To provide a global view of our design, we provide a detailed pseudocode here for clarity and improve the reproductivity.

---

**Algorithm 1:** Fisale Pipeline for Tow-Way FSI Problems

**Input:** Observations of fluid $\mathbf{u}_f$, solid $\mathbf{u}_s$, and interface $\mathbf{u}_b$ at time $t$

**Output:** Predicted states $\hat{\mathbf{u}}_f, \hat{\mathbf{u}}_s, \hat{\mathbf{u}}_b$ at $t + \Delta t$

**for** $h = 1$ **to** $H$ **do**

    Initialize latent ALE grid $\mathbf{g}_a^{(0,h)}$ via geometry-aware offset and $k$-NN edge set $E^{(h)}$

    Embed input features: $\mathbf{x}_f^{(0,h)} \leftarrow \text{Linear}(\mathbf{u}_f), \mathbf{x}_s^{(0,h)} \leftarrow \text{Linear}(\mathbf{u}_s), \mathbf{x}_b^{(0,h)} \leftarrow \text{Linear}(\mathbf{u}_b)$

**for** $l = 1$ **to** $L$ **do**

    **for** $h = 1$ **to** $H$ **do**

        **Encode physical quantities onto grid:**

        $\mathbf{p}_f \leftarrow \text{Encode}(\mathbf{x}_f^{(l-1,h)}, \mathbf{g}_a^{(l-1,h)})$

        $\mathbf{p}_s \leftarrow \text{Encode}(\mathbf{x}_s^{(l-1,h)}, \mathbf{g}_a^{(l-1,h)})$

        $\mathbf{p}_b \leftarrow \text{Encode}(\mathbf{x}_b^{(l-1,h)}, \mathbf{g}_a^{(l-1,h)})$

        **Partitioned Coupling Module (PCM):**

            Update solid state: $\mathbf{p}_s', \mathbf{p}_b' \leftarrow \text{CrossAttention}(\mathbf{p}_s, \mathbf{p}_b, \mathbf{p}_f, \mathbf{g}_a)$

            Update grid: $\mathbf{g}_a' \leftarrow \text{LaplacianSmooth}(\mathbf{p}_s', \mathbf{p}_f, \mathbf{p}_b', \mathbf{g}_a, E)$

            Update fluid state: $\mathbf{p}_f', \mathbf{p}_b'' \leftarrow \text{CrossAttention}(\mathbf{p}_f, \mathbf{p}_b', \mathbf{p}_s', \mathbf{g}_a')$

            Update interface influence: $\mathbf{p}_s'', \mathbf{p}_f'', \mathbf{p}_b''' \leftarrow \text{SelfAttention}(\mathbf{p}_s', \mathbf{p}_f', \mathbf{p}_b'', \mathbf{g}_a')$

        **Decode features:**

        $\mathbf{x}_f^{(l,h)} \leftarrow \text{Decode}(\mathbf{p}_f'', \mathbf{g}_a')$

        $\mathbf{x}_s^{(l,h)} \leftarrow \text{Decode}(\mathbf{p}_s'', \mathbf{g}_a')$

        $\mathbf{x}_b^{(l,h)} \leftarrow \text{Decode}(\mathbf{p}_b''', \mathbf{g}_a')$

    **Aggregate across scales:**

    $\mathbf{x}_f^{(l)} \leftarrow \text{FFN}(\text{Concat}_{h=1}^{H} \mathbf{x}_f^{(l,h)})$

    $\mathbf{x}_s^{(l)} \leftarrow \text{FFN}(\text{Concat}_{h=1}^{H} \mathbf{x}_s^{(l,h)})$

    $\mathbf{x}_b^{(l)} \leftarrow \text{FFN}(\text{Concat}_{h=1}^{H} \mathbf{x}_b^{(l,h)})$

    **if** $l < L$ **then**

        **Chunk fused features back to $H$ pathways:**

        $\{\mathbf{x}_f^{(l,h)}\}_{h=1}^{H} \leftarrow \text{Chunk}(\mathbf{x}_f^{(l)})$

        $\{\mathbf{x}_s^{(l,h)}\}_{h=1}^{H} \leftarrow \text{Chunk}(\mathbf{x}_s^{(l)})$

        $\{\mathbf{x}_b^{(l,h)}\}_{h=1}^{H} \leftarrow \text{Chunk}(\mathbf{x}_b^{(l)})$

**Output predictions:**

$\hat{\mathbf{u}}_f, \hat{\mathbf{u}}_s, \hat{\mathbf{u}}_b \leftarrow \text{Linear}(\mathbf{x}_f^{(L)}, \mathbf{x}_s^{(L)}, \mathbf{x}_b^{(L)})$

---

## B    DATASET

We evaluate our models in three public and curated datasets, whose information is summarized in Table 1. Note that these benchmarks involve the following three types of fluid-solid interaction tasks, which are widely explored in studies that focus only on fluid or solid:

- **Single-Step Prediction** (Li et al., 2021; Rahman et al., 2024; Li et al., 2025): Given a solution sequence $\{\mathbf{u}_0, \mathbf{u}_1, \ldots, \mathbf{u}_T\}$ of a time-dependent PDE, the goal is to learn a model $\mathcal{F}_\theta$ that maps the current state to the target state:

$$\hat{\mathbf{u}}_{t+\Delta t} = \mathcal{F}_\theta(\mathbf{u}_t)$$

where $\Delta t$ spans the next few time steps. During both training and inference, the model always receives the ground truth $\mathbf{u}_t$ as input to predict $\mathbf{u}_{t+\Delta t}$. This task is fundamental for learning local temporal dynamics and serves as a building block for simulating physical processes.

- **Autoregressive Simulation** (Pfaff et al., 2020; Sanchez-Gonzalez et al., 2020; Ma et al., 2024): Similar to single-step prediction, but during inference, the model recursively uses its own previous prediction as input except for the first step:

$$\hat{\mathbf{u}}_{t+1} = \mathcal{F}_\theta(\mathbf{u}_t), \quad \hat{\mathbf{u}}_{t+2} = \mathcal{F}_\theta(\hat{\mathbf{u}}_{t+1}), \quad \ldots \quad , \hat{\mathbf{u}}_{t+T} = \mathcal{F}_\theta(\hat{\mathbf{u}}_{t+T-1})$$

This method allows for long-term rollout of PDE solutions and is widely used in physics-informed forecasting and control.

- **Steady-State Inference** (Wu et al., 2024; Deng et al., 2024; Li et al., 2023d): Given a set of problem parameters $\lambda$ (e.g., environment conditions, boundary conditions, material properties), the objective is to learn a mapping directly from input parameters to the steady-state solution $\mathbf{u}^*$ of the PDE:

$$\mathcal{L}(\mathbf{u}^*, \lambda) = 0, \quad \mathbf{u}^* = \mathcal{F}_\theta(\lambda)$$

This parameter-to-solution formulation maps input conditions to the system's equilibrium state and plays a central role in engineering and scientific design problems concerned with steady-state behavior.

### B.1    STRUCTURE OSCILLATION

The structure oscillation problem, also known as "FLUSTRUK-A", is a well-established benchmark in the field of computational FSI. It models the interaction between an incompressible, viscous fluid and a thin, elastic beam attached to the rear of a rigid cylinder placed in a channel. The fluid flow around the cylinder induces unsteady forces on the beam, which, in turn, leads to self-sustained oscillations of the structure. These oscillations are periodic and strongly depend on the material properties of the beam, such as its density, elasticity, and damping (Turek & Hron, 2006). This problem plays a crucial role in validating and comparing the performance of FSI solvers due to its nonlinear, coupled nature. Unlike purely fluid or solid benchmarks, FLUSTRUK-A tests a solver's ability to accurately capture dynamic feedback between two physical domains. It is widely used in academic research and engineering applications, especially in domains where flow-induced vibrations (FIV) are significant, such as aerospace, civil engineering, and biomedical simulations (e.g., modeling blood flow through flexible vessels). The problem is particularly challenging due to the fine balance required between numerical stability and physical fidelity, making it an ideal dataset for developing and benchmarking advanced data-driven or physics-based models (Hoffman et al., 2012).

The dataset used in our experiments is proposed in CoDA-NO (Rahman et al., 2024). The computational domain is a two-dimensional channel of length 2.5 and height 0.41, containing a fixed circular cylinder of radius 0.05 centered at (0.2, 0.2), and a thin elastic beam attached to the rear of the cylinder with a length of 0.35 and thickness of 0.02. The fluid is modeled as water with a constant density of $1000 kg/m^3$. The flow enters the domain through the left boundary with a time-dependent fourth-order polynomial velocity profile that vanishes at the top and bottom walls. The inlet conditions vary across 28 predefined configurations, and the peak inlet velocity reaches approximately $4\ m/s$, enabling diverse and realistic flow conditions for each viscosity setting. The outlet (right boundary) applies a zero-pressure condition, and no-slip boundary conditions are enforced on the channel walls, the cylinder, and the elastic beam. To investigate different flow regimes, the dataset includes simulations with four viscosity values: $\mu \in \{0.5, 1, 5, 10\}$, resulting in Reynolds numbers approximately ranging from 4000 (for $\mu = 0.5$) to 200 (for $\mu = 10$). For the solid, the density is set to $1000\ kg/m^3$ with Lamé parameters $\lambda = 4.0 \times 10^6$ and $\mu = 2.0 \times 10^6$. Simulations are run up to a final time $T_f = 10$ seconds, using a fixed time step of $\delta t = 0.01$, resulting in 1000 time steps per trajectory. Samples share same physical domain and mesh. Each sample contains 1317 mesh points.

In our experiments, we set the prediction interval to $\Delta t = 4\delta t$ in order to allow the oscillation to evolve sufficiently and evaluate the model's ability to predict over longer time horizons. We first conduct training and evaluation on data with Reynolds numbers $Re \in \{200, 400, 2000\}$. The data frames is randomly split into training, validation, and test sets in a ratio of 8:1:1, resulting in 9561 training samples, 1195 validation samples, and 1196 test samples. We further test the out-of-distribution (OOD) generalization of trained models on a separate set of 498 samples generated with $Re = 4000$. This setup is similar in spirit to that used in CoDA-NO (Rahman et al., 2024), but with important differences. Specifically, CoDA-NO employs pretraining on the first 700 frames of each trajectory followed by few-shot fine-tuning. In contrast, our study does not involve any pretraining or fine-tuning procedures. All models are trained and evaluated under the same conditions, using longer time interval and randomly shuffled samples. This design ensures a fair comparison of in-distribution and OOD performance across models. Additionally, we use a different evaluation metric from CoDA-NO. The rationale for this choice is detailed in Appendix C.

## B.2 VENOUS VALVE

The venous valve problem models the dynamics of valve leaflets within veins, which play a critical role in ensuring unidirectional blood flow and preventing backflow in the human circulatory system (Bazigou & Makinen, 2013; Enderle & Bronzino, 2012). The system involves a thin, flexible leaflet that interacts with a pulsatile, incompressible fluid under physiological conditions. The valve opens and closes in response to changes in local pressure and flow rate, mimicking the behavior observed in venous circulation. This problem presents several unique challenges. First, the contact between the leaflets during valve closure introduces discontinuities and non-smooth behavior in the fluid-solid interface. Second, the large deformation of the leaflet requires robust modeling of nonlinear elasticity. Third, the highly transient, time-dependent nature of the flow, driven by periodic inlet conditions, demands accurate and stable single-step prediction to effectively simulate the full valve cycle. Due to these complexities, the venous valve problem serves as a stringent benchmark for evaluating FSI solvers, especially those aiming to operate under realistic biomedical conditions. It has important implications in biomedical research and healthcare applications, including the study of venous insufficiency, the design of prosthetic valves, and the development of patient-specific simulation tools for diagnosis.

To deeply investigate this problem and evaluate the effectiveness of Fisale, we constructed a simulation model based on related literature (Lin et al., 2023; Wang et al., 2025; Tikhomolova et al., 2020). The simulation is implemented using COMSOL Multiphysics (Multiphysics, 1998), a widely used finite element solver for coupled multiphysics problems. The domain shape of venous valve (Joda et al., 2016) is illustrated in Figure 5.

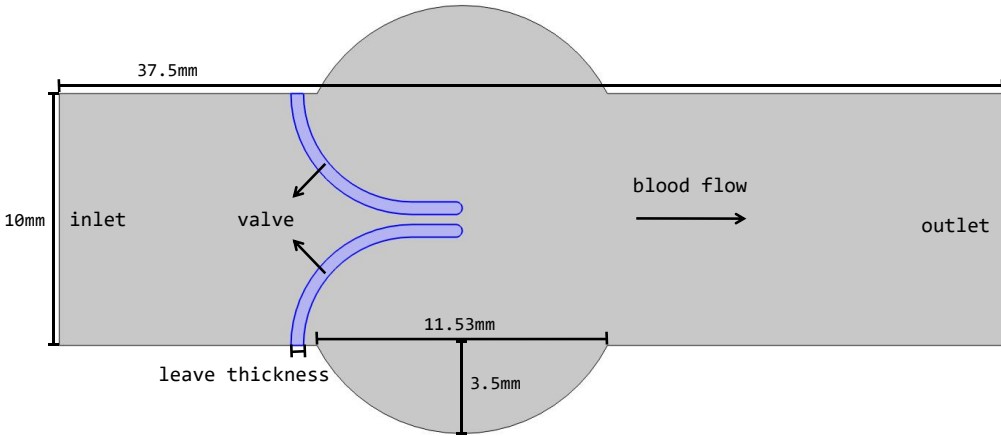

Figure 5: The physical domain of venous valve simulation model.

To simulate venous valve dynamics across a range of physiological variations, we define parameter sets for leaflet thickness, inlet blood velocity, and the mechanical properties of the valve tissue. These parameters are varied systematically using a full factorial combination scheme, allowing us to generate data that reflect a diverse set of biological scenarios corresponding to different genders,

age groups, and biological states. While some parameters and combinations may extend beyond typical values observed in healthy individuals, they are intentionally included to explore extreme or pathological scenarios. This broader coverage is important for studying disease-related valve dysfunction as well as for informing the design and testing of prosthetic valves. The specific parameter settings used in the simulations are summarized in Table 6.

Table 6: Parameter settings of venous valve simulation model.

| Leaflet Thickness (mm) | Inlet Blood Velocity (m/s) | Valve Material | |
| --- | --- | --- | --- |
| | | $C_1$ (MPa) | $C_2$ (MPa) |
| range(0.5,1.0,0.1) | range(0.1,0.6,0.1) | {0.01,0.05,0.1,0.15,0.2} | {0.001,0.005,0.01,0.02,0.05} |

The leaflet is modeled as a hyperelastic Mooney–Rivlin material (Boulanger & Hayes, 2001), which is widely used for soft biological tissues. The material behavior is governed by two coefficients, $C_1$ and $C_2$, representing the elastic response under deformation. The leaflet thickness is varied from $0.5mm$ to $1.0mm$ in increments of $0.1mm$, while the inlet blood velocity ranges from $0.1m/s$ to $0.6m/s$ with the same step size. The outlet applies a zero-pressure condition, and no-slip boundary conditions are enforced on the vessel walls. The blood is modeled as an incompressible Newtonian fluid, with density $\rho = 1050kg/m^3$ and dynamic viscosity $0.0035Pa \cdot s$. We formulate it as a transient FSI problem. A time-dependent inlet velocity function $\sin^2 \pi t$ is applied, modeling a periodic blood flow cycle with a period of 1 second. Simulations are performed with a time step of $0.01s$, resulting in 101 frames per trajectory. Each frame records multiple physical quantities, including current geometry, stress, pressure, and velocity. Based on the full factorial combination of parameter settings, we generate a total of 900 simulation trajectories. Among them, 720 trajectories are used for training, 90 for validation, and 90 for testing. Each sample contains 1693 mesh points on average.

## B.3 FLEXIBLE WING

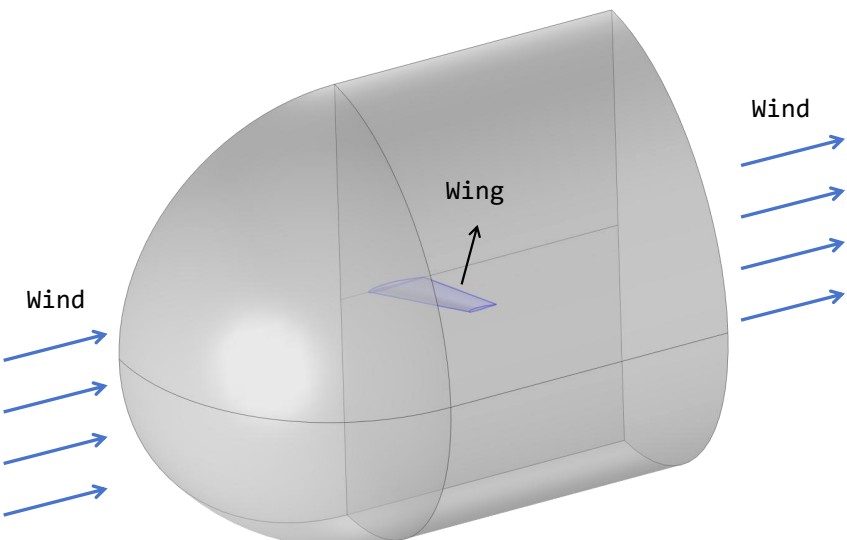

Figure 6: The physical domain of flexible wing simulation model.

The flexible wing problem is of practical importance in numerous engineering and biological contexts. In aerospace engineering, understanding the aerodynamics of flexible wings is crucial for designing next-generation aircraft, drones, and morphing airframes that can adapt to changing flight conditions. To better capture realistic aerodynamic behavior, we study a FSI problem involving a flexible wing. Unlike traditional rigid-wing models that assume a fixed structural geometry and static fluid–solid interface, the flexible wing can deform under aerodynamic loading, introducing strong two-way

coupling between the flow field and the structure. This coupling results in highly nonlinear and geometry-dependent behavior, including large elastic deformations and nontrivial load pressure across the wing surface. In this task, the goal is steady-state inference: given boundary conditions such as inflow velocity and structural material properties, the model aims to predict the equilibrium configuration of the wing and the corresponding steady-state fluid flow around it. This differs fundamentally from time-dependent FSI tasks in that it seeks a converged static solution representing the long-time behavior of the system, rather than predicting transient trajectories.

We follow the conventions (Bonnet et al., 2022; Valencia et al., 2025) to generate the dataset. The simulation domain is shown in Figure 6. The free-stream kinematic viscosity is $1.5 \times 10^{-5} m^2/s$. The wing sections are 24XX NACA airfoils with constant relative thickness (indicated by XX). The wing has a length of $1.5m$ and a root chord length of $1m$. The geometric parameters of the wing vary across samples include the relative thickness, the taper ratio (ratio between the tip and root chords), the sweep angle (angle between the quarter-chord line and a line perpendicular to the wing root). Additionally, the free-stream velocity, attack angle and the wing materials are also variations for diversity. These parameters are listed in Table 7.

Table 7: Parameter settings of flexible wing.

| Thickness | Taper Ratio | Sweep Angle (°) | Free-stream Velocity (m/s) | Attack Angle (°) | Wing Material |
|---|---|---|---|---|---|
| {10,12,14,16} | {0.5,0.6,0.7} | {10,25,40} | {50,75,100} | {-12,10,14,18} | Carbon Fibers Titanium Alloy Al-Zn-Mg Alloy |

Here, three different materials namely Carbon Fibers, Alpha-Beta Titanium Alloy and Al-Zn-Mg Alloy have been utilized based on previous literature (Chakraborty & Ghosh, 2022), which are widely used materials for air-wing analysis. The detailed material parameters are listed in Table 8.

Table 8: Material parameters of wings.

| Material Name | Density ($kg/m^3$) | Young's Modulus ($MPa$) | Poisson's Ratio |
|---|---|---|---|
| Carbon Fibers | $2.30 \times 10^{-6}$ | $2.30 \times 10^5$ | 0.210 |
| Alpha-Beta Titanium Alloy | $4.43 \times 10^{-6}$ | $1.13 \times 10^5$ | 0.342 |
| Al-Zn-Mg Alloy | $2.83 \times 10^{-6}$ | $7.20 \times 10^4$ | 0.327 |

The dataset samples are generated through a full factorial combination of the simulation parameters, resulting in a total of 1296 unique configurations. Among them, 1036 samples are used for training, 129 for validation, and 131 for testing. Each sample contains 37441 mesh points on average.

## C METRICS

We employ different metrics for specific tasks, adhering to the evaluation approaches in related works.

**Single-Step Prediction & Steady-State Inference: Relative L2** In line with prior studies on single-step prediction (Li et al., 2021; Rahman et al., 2024; Li et al., 2025) and steady-state inference (Wu et al., 2024; Deng et al., 2024; Li et al., 2023d) tasks, we use the relative L2 to assess performance. Given the input physical quantities $\mathbf{u}$ and the predictions $\hat{\mathbf{u}}$, the relative L2 is computed as:

$$\text{Relative L2} = \frac{\|\mathbf{u} - \hat{\mathbf{u}}\|}{\|\mathbf{u}\|}$$

It is worth noting that for the structure oscillation task, we adopt the Relative L2 error as the evaluation metric, instead of the Mean Square Error (MSE) used in CoDA-NO (Rahman et al., 2024). This

task involves predicting multiple physical quantities with different units and orders of magnitude. While CoDA-NO reported MSE results based on normalized data to mitigate the influence of scale differences, this approach may not fully reflect model performance on the original data distribution. In contrast, Relative L2 is a more commonly used metric for this task. By computing the Relative L2 error for each physical quantity separately and then averaging the results, we effectively account for differences in scale while preserving fidelity to the original, unnormalized data.

**Autoregressive Simulation: RMSE** Consistent with works (Pfaff et al., 2020; Sanchez-Gonzalez et al., 2020; Ma et al., 2024) focused on autoregressive simulation tasks, we use Root Mean Square Error (RMSE) as the evaluation metric. Given the input physical quantities $\mathbf{u}$ and the predictions $\hat{\mathbf{u}}$, RMSE is calculated as:

$$\text{RMSE} = \sqrt{\frac{1}{N} \sum_{i=1}^{N} \|\mathbf{u}_i - \hat{\mathbf{u}}_i\|^2}$$

During training, we compute the loss using normalized data, which is used for backpropagation and parameter updates. In the evaluation and test phases, we report the RMSE for each physical quantity within each domain based on the original data.

## D IMPLEMENTATION

As shown in Table 9, Fisale and all baseline models are trained and tested using the same training strategy. We utilize relative L2 as loss function for single-step prediction and steady-state inference tasks, and RMSE for autoregressive simulation task. For different physical domains, we add each loss with equal weights. To ensure fair comparisons, we first approximate the parameter count of all baselines to match that of Fisale and then adjust their parameters to minimize overfitting and achieve better performance. All experiments are conducted on a single RTX 3090 GPU (24GB memory) and repeated three times. We provide the parameter count of each model in Table 10 for reference.

Table 9: Training and Model Configurations of Fisale. The definition of batch size differs between autoregressive simulation task and other two tasks. For single-step prediction and steady-state inference tasks, the batch size refers to the number of samples in a batch. For autoregressive simulation, only one sample is processed during each forward and backward pass, and the batch size corresponds to the number of time steps in the sample. Since GPU memory usage varies across different models, the batch sizes of baseline models in the autoregressive simulation task are dynamically adjusted to avoid GPU memory overflow while maintaining performance.

| Datasets | Training Configuration | | | Model Configuration | | | |
| --- | --- | --- | --- | --- | --- | --- | --- |
| | Epochs | LR | Batch | Level ($L$) | Pathway ($H$) | Grid Shape ($M$) | Channels ($D$) |
| Structure Oscillation | 100 | $1 \times 10^{-3}$ | 50 | 2 | 2 | $[16, 16]$ $[8, 8]$ | $[64, 64]$ |
| Venous Valve | 100 | $1 \times 10^{-3}$ | 50 | 2 | 2 | $[16, 16]$ $[8, 8]$ | $[64, 64]$ |
| Flexible Wing | 100 | $5 \times 10^{-4}$ | 1 | 3 | 2 | $[5, 5, 5]$ $[4, 4, 4]$ | $[96, 128]$ |

We implement baselines based on official and popular implementations. For autoregressive simulation task, we uniformly add noise with a mean of 0 and a variance of 0.001 to improve the error accumulation control during rollout. Since several neural operators and transformer-based baselines are primarily designed for fluid scenarios with Eulerian settings, they are not naturally suited for handling scenarios with Lagrangian views, such as two-way FSI problems. We preprocess the data to adapt it for use with these baselines. Specifically, we first align the feature dimensions of different domains using padding, then concatenate along the length dimension to combine all domains into a single large sequence of points. For models that struggle to handle the Lagrangian setting (Rahman et al., 2024; Li et al., 2023c; Wu et al., 2023), we map each point to a regular grid, transforming the data into an Eulerian representation. Specifically, for a point sequence of size $N \times D$, we discretize a

Table 10: Parameter count of baseline models and Fisale.

|            | Structure Oscillation | Venous Valve | Flexible Wing |
|------------|:---------------------:|:------------:|:-------------:|
| Geo-FNO    | 1.60 M                | 1.60 M       | 5.22 M        |
| GINO       | 1.72 M                | /            | 5.17 M        |
| LSM        | 2.23 M                | 2.23 M       | /             |
| CoDANO     | 1.83 M                | 1.83 M       | 5.01 M        |
| LNO        | 1.62 M                | /            | 5.34 M        |
| Galerkin   | 1.74 M                | 1.74 M       | 3.18 M        |
| GNOT       | 1.64 M                | 1.64 M       | 4.64 M        |
| ONO        | 1.63 M                | /            | 5.07 M        |
| Transolver | 1.55 M                | 1.55 M       | 4.61 M        |
| MGN        | 1.82 M                | 1.82 M       | 4.86 M        |
| HOOD       | 1.79 M                | 1.79 M       | 5.31 M        |
| AMG        | 1.34 M                | 1.34 M       | 4.71 M        |
| Fisale     | 1.54 M                | 1.54 M       | 4.88 M        |

cubic space $[x, y, z] \in [-1, 1]^3$ into a regular grid, with the number of discretization points along each axis set to $\lceil \sqrt[3]{N} \rceil$. We then concatenate each point's coordinates and physical quantities with the corresponding grid points based on their matching order. Excess grid cells are padded to align dimensions. For other baseline models, we utilize geometry transformation functions provided in their implementations. Additionally, the fixed boundary condition is maintained by directly setting the displacement of the corresponding region as zero like the operation in MGN Pfaff et al. (2020).

**Efficiency** We report GPU memory usage based on measurements from the operating system, which include memory pre-allocated by PyTorch's caching allocator. Although not all of this memory is actively used at every moment, it remains unavailable to other processes once reserved. Therefore, this reporting strategy offers a conservative yet practical estimate of the actual hardware demands during training. It better reflects the real-world resource constraints typically encountered in deployment scenarios. To ensure fairness, all models are evaluated using PyTorch's default memory management on a single RTX 3090 GPU.

## E  STANDARD DEVIATION

Table 11: Standard deviations on Structure Ocsillation experiment.

|            | Solid ($\times10^{-3}$) | Fluid ($\times10^{-2}$) | Interface ($\times10^{-2}$) |
|------------|:-----------------------:|:-----------------------:|:---------------------------:|
| Geo-FNO    | $\pm0.01$               | $\pm0.05$               | $\pm0.01$                   |
| GINO       | $\pm0.50$               | $\pm2.02$               | $\pm0.78$                   |
| LSM        | $\pm0.01$               | $\pm0.72$               | $\pm0.02$                   |
| CoDANO     | $\pm0.07$               | $\pm0.05$               | $\pm0.07$                   |
| LNO        | $\pm0.03$               | $\pm0.29$               | $\pm0.12$                   |
| Galerkin   | $\pm0.08$               | $\pm0.03$               | $\pm0.02$                   |
| GNOT       | $\pm0.02$               | $\pm0.05$               | $\pm0.06$                   |
| ONO        | $\pm0.24$               | $\pm0.92$               | $\pm0.12$                   |
| Transolver | $\pm0.07$               | $\pm0.05$               | $\pm0.08$                   |
| MGN        | $\pm0.04$               | $\pm0.03$               | $\pm0.03$                   |
| HOOD       | $\pm0.05$               | $\pm0.03$               | $\pm0.04$                   |
| AMG        | $\pm0.01$               | $\pm0.01$               | $\pm0.01$                   |
| Fisale     | $\pm0.01$               | $\pm0.04$               | $\pm0.01$                   |

We repeat all the experiments three times and provide standard deviations here in Table 11, 12 and 13. It is worth noting that autoregressive simulation involves a rollout process based on sequential

predictions, where error accumulation is inevitable. When the rollout trajectory is long, such accumulated errors can become unstable and difficult to control. In addition, our venous valve experiment poses a greater challenge to result stability due to the presence of multiple target physical quantities across different physical domains. As a result, this task exhibits higher variance compared to the other two tasks. Moreover, to ensure fairness and maintain the integrity of each method, we adopt the initialization schemes provided in the official and popular implementations, rather than applying a unified initialization across all baselines. As a result, the scale and variance of network parameters vary across models, which may lead to differences in performance variance. As shown in Table 12, although Fisale does not always exhibit the lowest variance across all physical quantities, its overall performance remains consistently stable. This indicates that Fisale not only achieves superior performance, but also maintains robustness in long-time rollout, which is an essential property for reliable simulation in complex FSI scenarios.

Table 12: Standard deviations on Venous Valve experiment.

| | Solid | | Fluid | | Interface | | |
| --- | --- | --- | --- | --- | --- | --- | --- |
| | Geometry $(\times 10^{-2})$ | Stress $(\times 10^{0})$ | Pressure $(\times 10^{0})$ | Velocity $(x)$ $(\times 10^{-2})$ | Geometry $(\times 10^{-2})$ | Stress $(\times 10^{0})$ | Pressure $(\times 10^{0})$ |
| Geo-FNO | $\pm 3.88$ | $\pm 319.62$ | $\pm 12.61$ | $\pm 1.69$ | $\pm 4.19$ | $\pm 483.77$ | $\pm 11.81$ |
| LSM | $\pm 4.26$ | $\pm 216.70$ | $\pm 13.08$ | $\pm 2.12$ | $\pm 4.56$ | $\pm 286.48$ | $\pm 11.13$ |
| CoDANO | $\pm 6.67$ | $\pm 552.81$ | $\pm 18.19$ | $\pm 2.87$ | $\pm 6.30$ | $\pm 603.29$ | $\pm 16.16$ |
| Galerkin | $\pm 0.71$ | $\pm 79.86$ | $\pm 4.24$ | $\pm 0.86$ | $\pm 1.02$ | $\pm 107.30$ | $\pm 2.62$ |
| GNOT | $\pm 1.66$ | $\pm 258.56$ | $\pm 8.69$ | $\pm 1.04$ | $\pm 1.89$ | $\pm 236.67$ | $\pm 3.87$ |
| Transolver | $\pm 2.18$ | $\pm 176.98$ | $\pm 3.37$ | $\pm 0.81$ | $\pm 2.50$ | $\pm 184.31$ | $\pm 4.23$ |
| MGN | $\pm 6.28$ | $\pm 417.58$ | $\pm 15.00$ | $\pm 2.21$ | $\pm 5.46$ | $\pm 448.81$ | $\pm 9.54$ |
| HOOD | $\pm 4.50$ | $\pm 359.47$ | $\pm 13.37$ | $\pm 1.48$ | $\pm 3.95$ | $\pm 351.29$ | $\pm 7.29$ |
| AMG | $\pm 5.28$ | $\pm 329.23$ | $\pm 8.47$ | $\pm 1.37$ | $\pm 3.18$ | $\pm 313.47$ | $\pm 7.36$ |
| Fisale | $\pm 2.01$ | $\pm 189.26$ | $\pm 3.05$ | $\pm 0.92$ | $\pm 1.92$ | $\pm 223.60$ | $\pm 3.58$ |

Table 13: Standard deviations on Flexible Wing experiment. Some baselines are not included due to severe degradation, where standard deviations are too unstable to provide meaningful reference.

| | Solid $(\times 10^{-2})$ | Fluid $(\times 10^{-2})$ | Interface $(\times 10^{-2})$ |
| --- | --- | --- | --- |
| Geo-FNO | $\pm 0.01$ | $\pm 0.04$ | $\pm 0.24$ |
| CoDANO | $\pm 0.18$ | $\pm 0.15$ | $\pm 0.27$ |
| LNO | $\pm 0.02$ | $\pm 0.02$ | $\pm 0.04$ |
| Galerkin | $\pm 0.09$ | $\pm 0.19$ | $\pm 0.25$ |
| GNOT | $\pm 0.02$ | $\pm 0.04$ | $\pm 0.01$ |
| Transolver | $\pm 0.01$ | $\pm 0.04$ | $\pm 0.03$ |
| MGN | $\pm 0.01$ | $\pm 0.02$ | $\pm 0.01$ |
| HOOD | $\pm 0.01$ | $\pm 0.02$ | $\pm 0.02$ |
| Fisale | $\pm 0.01$ | $\pm 0.01$ | $\pm 0.02$ |

# F  EFFICIENCY

Efficiency is a key concern in practical applications, especially in large-scale scenarios involving massive mesh points. Therefore, we provide a dedicated discussion on the computational efficiency of Fisale here.

In the computation pipeline of Fisale, the major modules include ALE initialization, physical quantity encoding and decoding, attention-based Partitioned Coupling Module (PCM), and FFN-based aggregation.

- For ALE initialization, the most expensive step is computing offsets, which involves calculating distances between $N$ observed physical points and $M$ ALE grid points, resulting in a complexity of $\mathcal{O}(NM)$. After constructing the ALE grid, we apply k-Nearest Neighbors (kNN) to build edge connections, which incurs a complexity of $\mathcal{O}(M^2 D)$.
- In the encoding stage, attention-like weights and spatial mappings are computed, both with a complexity of $\mathcal{O}(NMD)$. Similarly, the decoding stage also involves mapping from grids to physical points, again costing $\mathcal{O}(MND)$.
- During the PCM evolution, we adopt linear attention, leading to a complexity of $\mathcal{O}(MD^2)$.
- The FFN-based aggregation module uses stacked linear layers over $N$ physical points, contributing a complexity of $\mathcal{O}(ND^2)$.

We have ignored constant factors such as level $L$, pathway $H$, problem dimension $C$ and scalar coefficients. The total computational complexity is:

$$\mathcal{O}(NMD + M^2 D + ND^2 + MD^2)$$

In this expression, $M$ (the number of ALE grid points) and $D$ (the feature channels) are user-defined hyperparameters, while $N$ (the number of physical observation points) is determined by the task and dataset scale. Typically, both $M \ll N$ and $D \ll N$. For instance, in the Flexible Wing task, $N > 10^4$, while in the other two tasks, $N > 10^3$. In contrast, $M$ and $D$ are both on the order of $10^2$. Therefore, we can approximate the overall computational complexity of Fisale as growing linearly with the problem size $N$, making it scalable to large-scale FSI scenarios. We further conduct an experiment to demonstrate the efficiency of Fisale.

Table 14: The efficiency comparison on Flexible Wing dataset with more than 35,000 mesh points for each sample in average. The Running Time is measured by the time to complete one epoch. Since the number of mesh points varies across samples, we report the peak GPU memory usage observed within a single epoch. Additionally, the recorded runtime includes the time spent on constructing graph edges. As a result, models like HOOD (Grigorev et al., 2023) and AMG (Li et al., 2025), which involve multiple rounds of dynamic edge construction during iteration steps, become particularly time-consuming.

| | Parameters (M) | Running Time (S) | GPU Memory (GiB) | Mean Relative L2 ($\downarrow$) |
|---|---|---|---|---|
| Geo-FNO | 5.22 | 71.97 | 0.66 | 0.0524 |
| GINO | 5.17 | 223.74 | 22.95 | 0.4745 |
| CoDANO | 5.01 | 1167.47 | 17.48 | 0.1096 |
| LNO | 5.34 | 50.4 | 1.60 | 0.0235 |
| Galerkin | 3.18 | 129.40 | 4.15 | 0.0577 |
| GNOT | 4.64 | 246.42 | 13.04 | 0.0289 |
| ONO | 5.07 | 152.87 | 7.46 | 0.2179 |
| Transolver | 4.61 | 245.75 | 11.09 | 0.0164 |
| MGN | 4.86 | 506.77 | 22.50 | 0.0202 |
| HOOD | 5.31 | $> 1500$ | 22.80 | 0.0191 |
| AMG | 4.71 | $> 1500$ | 21.92 | 0.2185 |
| Fisale | 4.88 | 296.30 | 3.10 | 0.0136 |

As shown in Table 14, Fisale achieves a favorable trade-off between efficiency and predictive performance. Benefiting from its parallel module design, Fisale maintains a relatively wide architecture under a similar parameter scale, which alleviates the memory burden associated with gradient storage during backpropagation. Moreover, by decomposing the physical domain into fluid, solid, and interface components, Fisale effectively splits large matrices into smaller submatrices. This reduces

the memory footprint of intermediate computations. Together, these design choices lead to significantly lower GPU memory usage compared to other attention-based, GNN-based models and most neural operators. Although Fisale involves kNN operations and multiple attention passes within each module, which limit its speed advantage, its runtime remains within a practically acceptable range. In summary, Fisale delivers high prediction accuracy with minimal memory overhead and without significant computational time increase, making it a good candidate for large-scale, real-world FSI applications.

# G ABLATION STUDY

Beyond the main results, we conduct a series of ablation studies to comprehensively evaluate the design. All ablation study experiments are conducted on the Flexible Wing task, which is a challenging long-range steady-state inference problem and contains more than 35,000 mesh points per instance.

**Explicit interface component** In our design, recognizing the importance of the coupling interface, we explicitly model it as a separate component on par with the solid and fluid. Here, we explore the necessity of this operation. Specifically, since the interface inherently shares properties (solid stress, fluid pressure, and velocity) from both the fluid and solid domains, we cannot directly merge it into one of them. Hence, we duplicate the interface coordinates: one copy is concatenated with the physical attributes of the solid and included as part of the solid input, while the other is concatenated with the attributes of the fluid and treated as part of the fluid input. For the output, we average the coordinates from the two branches as the final position of the interface and calculate the loss on each attribute. The network architecture remains unchanged except for removing the projection and deprojection of the interface component. The result is shown in Table 15. We can observe that removing the interface component leads to notable degraded performance. Since the behavior of the fluid-solid interface differs from both the solid and the fluid, merging it with either domain not only reduces the accuracy of the interface itself, but also adversely affects the evolution of the fluid and solid domains. This confirms that modeling the interface as an independent component contributes positively to the predictions.

Table 15: Ablation study of the explicit interface component.

|  | Solid | Fluid | Interface | Mean Relative L2 ($\downarrow$) | Decrease |
|---|---|---|---|---|---|
| Fisale | 0.0042 | 0.0155 | 0.0211 | 0.0136 | - |
| $w/o$ explicit interface | 0.0061 | 0.0212 | 0.0251 | 0.0175 | 28.68% |

**Ordering of the PCM** In fact, the Partitioned Coupling Algorithm is a flexible framework, and the specific ordering used in the our design is one representative choice rather than a fixed requirement. As we described in Section 2, a common and intuitive coupling loop (Placzek et al., 2009) proceeds as: the fluid exerts pressure on the solid, causing deformation; this alters the geometry and updates the grid positions; the updated grid affects the fluid state, which is then updated; finally, the updated fluid and solid states jointly determine the interface dynamics. Given this cycle, it is reasonable to begin with either the solid or the fluid, as both influence the evolution of the system. Different update sequences can be also physically justified. To further explore the order influence, we systematically test alternative update orders by permuting the four components. As shown in Table 16, we observe that: regardless of the specific update order, the model achieves comparable performance across all permutations. This indicates that the PCM is inherently a flexible and robust framework. Since Fisale adopts a stacked architecture, the update order of components does not need to remain fixed within each layer. Instead, the interactions among fluid, solid, grid, and interface can be iteratively adjusted through vertical information flow across layers. This iterative propagation helps compensate for local order choices, allowing the model to refine cross-domain interactions over multiple stages. Therefore, while we adopt a physically reasonable ordering in our implementation, the model's performance remains stable under other plausible orderings, further validating the flexibility of PCM.

**Replace PCM with a simpler attention module** We conduct an ablation experiment in which we replace the entire multi-stage attention cascade with a simpler module including an attention layer followed by an FFN. We concatenate four components (solid, fluid, interface, grid) as input like the

Table 16: Ablation study of the update ordering within PCM.

| Ordering | Solid | Fluid | Interface | Mean Relative L2 ($\downarrow$) |
|---|---|---|---|---|
| fluid-grid-solid-interface | 0.0043 | 0.0155 | 0.0206 | 0.0135 |
| grid-solid-fluid-interface | 0.0042 | 0.0155 | 0.0206 | 0.0134 |
| grid-solid-interface-fluid | 0.0041 | 0.0157 | 0.0216 | 0.0138 |
| grid-interface-solid-fluid | 0.0041 | 0.0155 | 0.0212 | 0.0136 |
| solid-fluid-interface-grid | 0.0044 | 0.0161 | 0.0213 | 0.0139 |
| solid-grid-fluid-interface | 0.0042 | 0.0155 | 0.0211 | 0.0136 |

representation format in learning-based fluid field and keep comparable model parameters by modify the latent dimension. As shown in the Table 17, the replacement leads to a drop in performance. This degradation is observed consistently across all regions. This indicates that cross-domain attentions in our design play a crucial role in modeling the intricate interactions among components. Although the simplified variant is more compact in structure, it lacks the ability to disentangle and coordinate domain-specific interactions, which are essential in FSI systems.

Table 17: Ablation study of replacing PCM with a simpler attention module.

| | Params (M) | Solid | Fluid | Interface | Mean Relative L2 ($\downarrow$) | Decrease |
|---|---|---|---|---|---|---|
| Fisale | 4.88 | 0.0042 | 0.0155 | 0.0211 | 0.0136 | - |
| Simpler Module | 4.96 | 0.0045 | 0.0178 | 0.0223 | 0.0149 | 9.56% |

**Multiscale Latent ALE Grids** To evaluate the effectiveness of multiscale design within Fisale, we perform two scaling experiments. First, we explore the effect of mesh resolution $M$. To isolate this factor, we use a single pathway and gradually increase $M$ to investigate how it influences model performance. Except for the $M$ and channel number, other settings are kept. The channel is set as 128 for each experiment. As shown in Table 18, increasing the resolution level $M$ in the single-pathway setting leads to slight improvements in performance. However, overall, the performance remains largely consistent across different values of $M$. This trend suggests that, in a single-resolution configuration, increasing resolution alone offers limited benefit. Each fixed-resolution mesh captures only a specific scale of geometric and physical patterns, and lacks the flexibility to simultaneously represent both coarse global structures and fine local details. Moreover, even when the parameter count increases to match that of our main model, the accuracy still lags behind the multi-resolution two-pathway design. This indicates that the exact value of $M$ is not a dominant critical factor in determining the model's learning capability.

Table 18: Scaling results of mesh resolution $M$ with a single pathway.

| $M^{1/3}$ | Params (M) | Solid | Fluid | Interface | Mean Relative L2 ($\downarrow$) |
|---|---|---|---|---|---|
| 3 | 2.48 | 0.0047 | 0.0171 | 0.0226 | 0.0148 |
| 5 | 2.63 | 0.0047 | 0.0169 | 0.0224 | 0.0147 |
| 7 | 3.66 | 0.0044 | 0.0171 | 0.0221 | 0.0145 |
| 8 | 5.11 | 0.0045 | 0.0172 | 0.0223 | 0.0147 |
| 9 | 7.81 | 0.0046 | 0.0167 | 0.0219 | 0.0144 |
| 10 | 12.50 | 0.0049 | 0.0172 | 0.0227 | 0.0149 |

Second, to further examine whether multi-resolution pathways can indeed capture richer and more complementary information, we conduct an additional ablation study on the high-resolution pathway $H$. The setting and results are shown in Table 19 and Figure 7, respectively.

Through the results we find that increasing the number of resolutions $H$ in the high-resolution pathway leads to clear performance gains. Notably, moving from $H = 1$ to $H = 2$ results in a

Table 19: Scaling settings of pathways $H$.

| $H$ | $M^{1/3}$ | Channels | Params (M) |
|---|---|---|---|
| 1 | [5] | [128] | 2.63 |
| 2 | [4, 5] | [128, 96] | 4.88 |
| 3 | [4, 5, 6] | [128, 96, 64] | 6.99 |
| 4 | [3, 4, 5, 6] | [128, 128, 96, 64] | 11.99 |
| 5 | [3, 4, 5, 6, 7] | [128, 128, 96, 64, 64] | 15.69 |

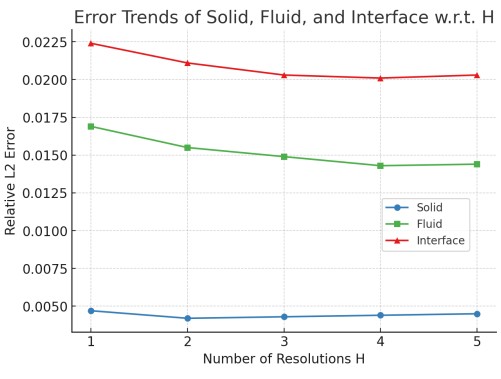

(a) Error trends of solid, fluid, and interface.

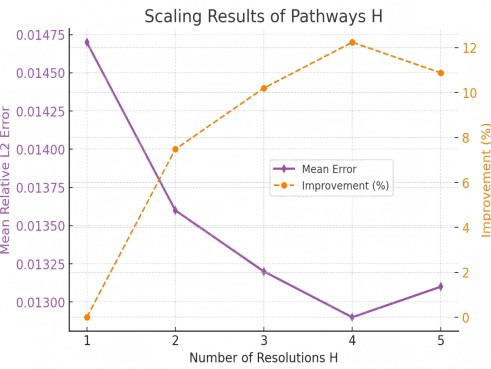

(b) Mean error and relative improvement.

Figure 7: Scaling results of pathways $H$.

significant improvement in overall accuracy demonstrating that the incorporation of multi-resolution features is highly beneficial for FSI tasks. This supports the idea that coupling information across multiple spatial scales is critical to capture complex cross-domain dynamics. As $H$ continues to increase, we observe gradual but diminishing improvements. This trend suggests two important insights:

- Multi-resolution aggregation is indeed effective, especially when moving from single-scale to dual- or tri-scale setups.

- Beyond a certain point (H = 4 for example), the model reaches a saturation limit, where adding more resolutions and parameters yields only marginal gains or even degradation. This is due to the limited data capacity or the task-specific resolution requirements already being fulfilled.

These findings highlight both the value and the limitations of increasing resolution diversity within the architecture. Our final selection of hyperparameters strikes a practical balance between performance and efficiency.

## H  SHOWCASES

### H.1  STRUCTURE OSCILLATION

Figures 8 and 9 present qualitative comparisons and corresponding error maps on the Structure Oscillation benchmark regarding the solid displacement, fluid $x$-velocity, and fluid pressure. This two-way FSI problem is characterized by strong mutual influence between the fluid and the solid: pressure from the fluid deforms the solid, while the resulting structural motion feeds back to alter the surrounding flow. Accurate prediction in such scenarios requires simultaneous fidelity in both the fluid and solid domains. As shown in Figures 8 and 9, several baseline models suffer from severe artifacts, including discontinuities in flow velocity and distorted solid shapes, which break the physical coupling and lead to unstable predictions. In contrast, Fisale maintains global coherence across the

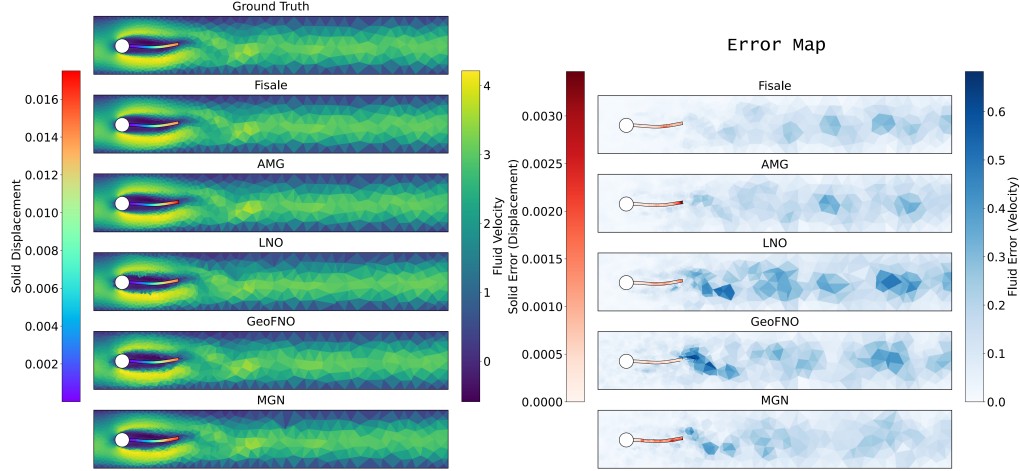

Figure 8: Showcase comparison of structure oscillation task. The solid deformation and fluid velocity are shown.

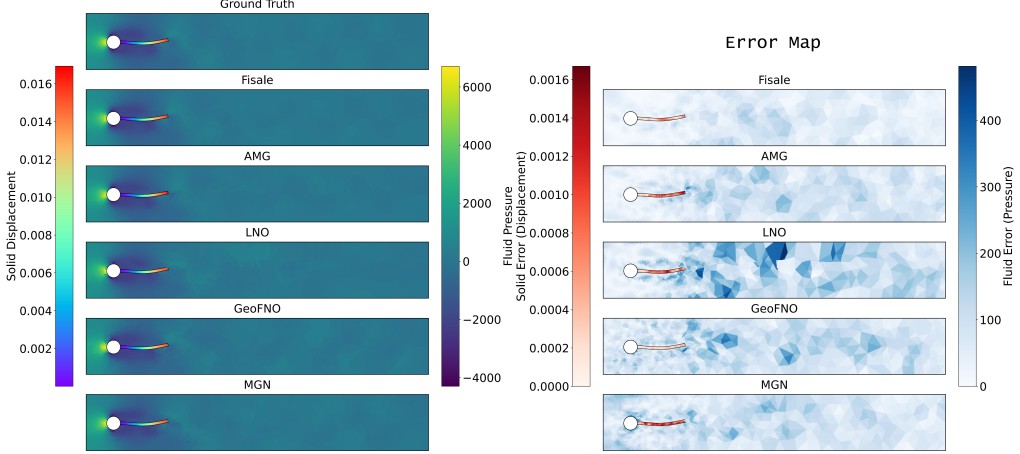

Figure 9: Showcase comparison of structure oscillation task. The solid deformation and fluid pressure are shown.

entire domain: the solid structure remains intact and physically plausible, and the surrounding flow field is smooth and consistent.

This consistency stems from Fisale's architectural design: the explicit modeling of the coupling interface ensures localized continuity across domains; the multiscale latent ALE representation provides a flexible and unified embedding for cross-domain geometries; and the stacked partitioned coupling modules enable progressive and iterative updates across the fluid–solid system. Together, these components allow Fisale to preserve the integrity of the coupling dynamics, leading to more reliable predictions in highly nonlinear, strongly coupled FSI regimes.

## H.2 VENOUS VALVE

The Table 20 contains the results of physical quantities that not included in main experimental table. Figure 10 and Figure 11 present qualitative comparisons and corresponding error maps for the venous valve task, focusing on valve deformation, stress, blood flow velocity, and pressure. In the early to mid stages of the rollout (before time step 55), Transolver is able to maintain the overall valve shape. However, the fluid field exhibits more significant error accumulation, with uneven distribution

Table 20: Performance comparison on Venous Valve. Supplemetary results of Table 4. We record RMSE-all, the average RMSE of the whole rollout trajectory and all samples. A smaller value indicates better performance.

| | Fluid | | Interface | |
| --- | --- | --- | --- | --- |
| | Geometry | Velocity ($y$) | Velocity ($x$) | Velocity ($y$) |
| Geo-FNO | 0.2850 | 0.0225 | 0.0780 | 0.0174 |
| LSM | 0.4022 | 0.0323 | 0.0825 | 0.0196 |
| CoDANO | 0.6203 | 0.0378 | 0.1042 | 0.0265 |
| Galerkin | 0.2811 | 0.0195 | 0.0658 | 0.0130 |
| GNOT | 0.2971 | 0.0182 | 0.0891 | 0.0157 |
| Transolver | 0.2867 | 0.0183 | 0.0641 | 0.0132 |
| MGN | 0.3851 | 0.0211 | 0.1018 | 0.0193 |
| HOOD | 0.3216 | 0.0204 | 0.0706 | 0.0152 |
| AMG | 0.3159 | 0.0193 | 0.0846 | 0.0161 |
| **Fisale** | **0.2337** | **0.0129** | **0.0542** | **0.0119** |

and particularly large errors near the valve opening. As the rollout progresses (after time step 65), the stability of Transolver degrades further, and noticeable geometric distortions appear in the valve structure. This can be attributed to its homogeneous modeling across the entire domain, which limits its ability to distinguish between fluid and solid regions and to capture their dynamic interactions at the interface. In contrast, Fisale explicitly models fluid, solid, and the coupling interface as separate components, assigning equal attention to each. This enables the model to better track interface dynamics and structural changes. As shown in Figure 10 and Figure 11, Fisale maintains valve shape stability even over long rollout trajectories and produces more accurate predictions for both fluid and solid fields, especially around the opening during peak states.

## H.3 FLEXIBLE WING

Figure 12 presents qualitative comparisons and corresponding error maps on the Flexible Wing dataset, focusing on wing deformation, stress, and surface pressure. The figure reveals clear spatial patterns across these physical quantities: deformation is concentrated at the tip of the wing, where the structural flexibility is highest; stress peaks at the root of the wing, where the wing connects to the fuselage; and pressure is distributed primarily along the wind-facing side of the wing. Such region-specific distribution behaviors introduce challenges for accurate prediction.

The figure also highlights the difficulty faced by models that apply a homogeneous modeling strategy across the entire domain. These models struggle to effectively capture the region-specific dynamics of each physical quantity. Moreover, in this dataset, the number of mesh points in the fluid domain significantly exceeds that in the solid domain, making it easy for solid-related information to be overwhelmed. Some baselines exhibit severe degradation, while others fail to distinguish fluid, solid, and surface regions, which hinders their ability to extract useful signals from dense mesh representations. This further demonstrates the effectiveness of Fisale's domain-aware design in capturing cross-domain dynamics and maintaining robustness under such challenging conditions.

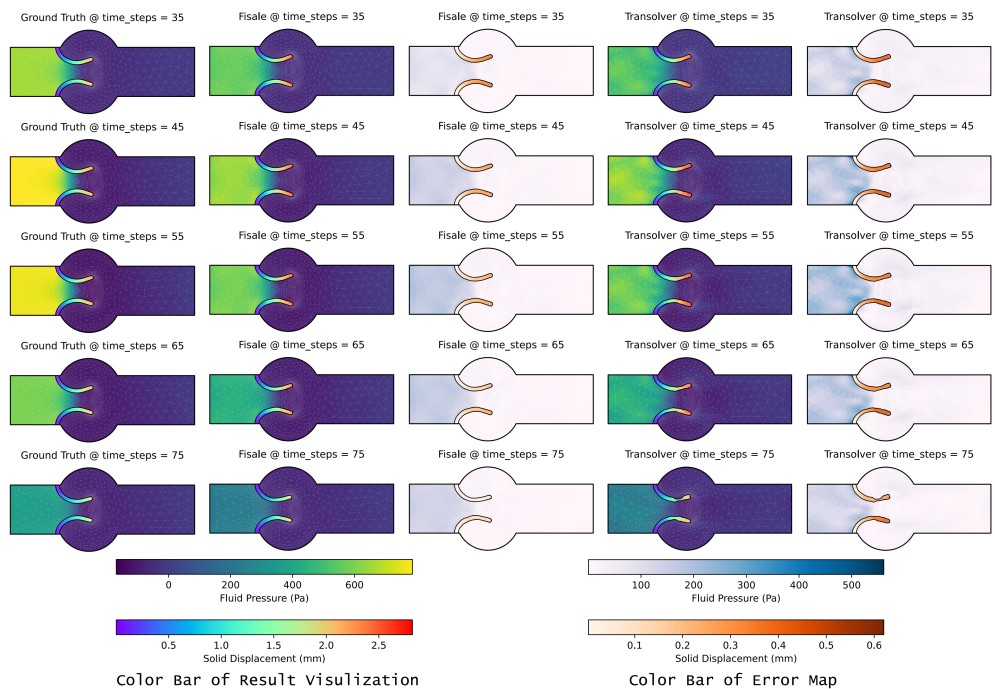

Figure 10: Showcase comparison of venous valve task. The solid deformation and fluid pressure are shown.

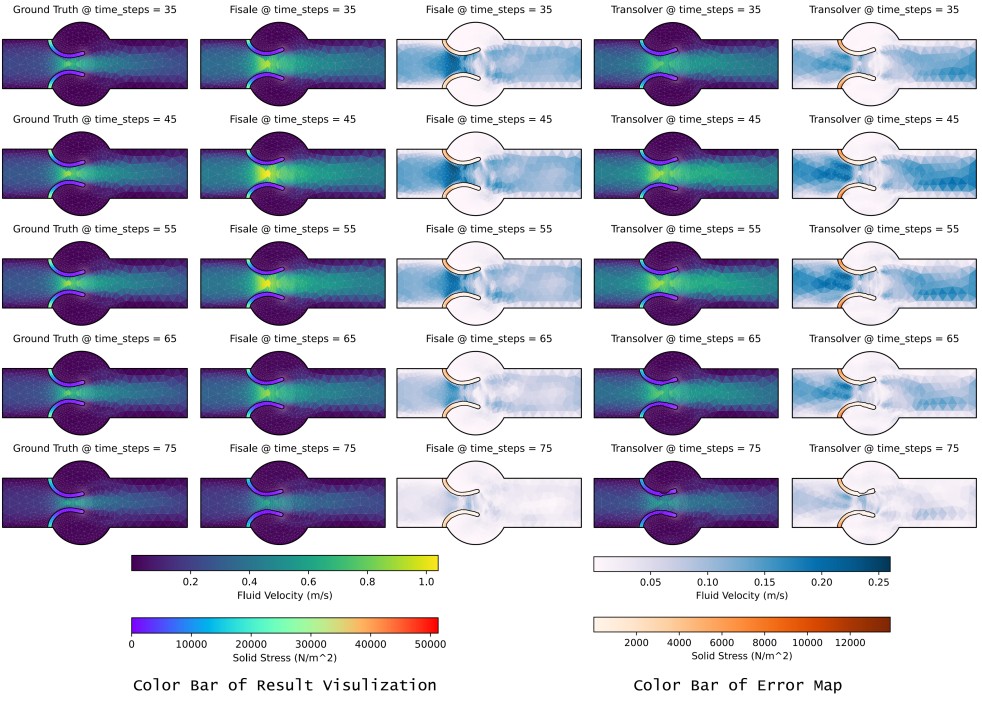

Figure 11: Showcase comparison of venous valve task. The solid stress and fluid velocity are shown.

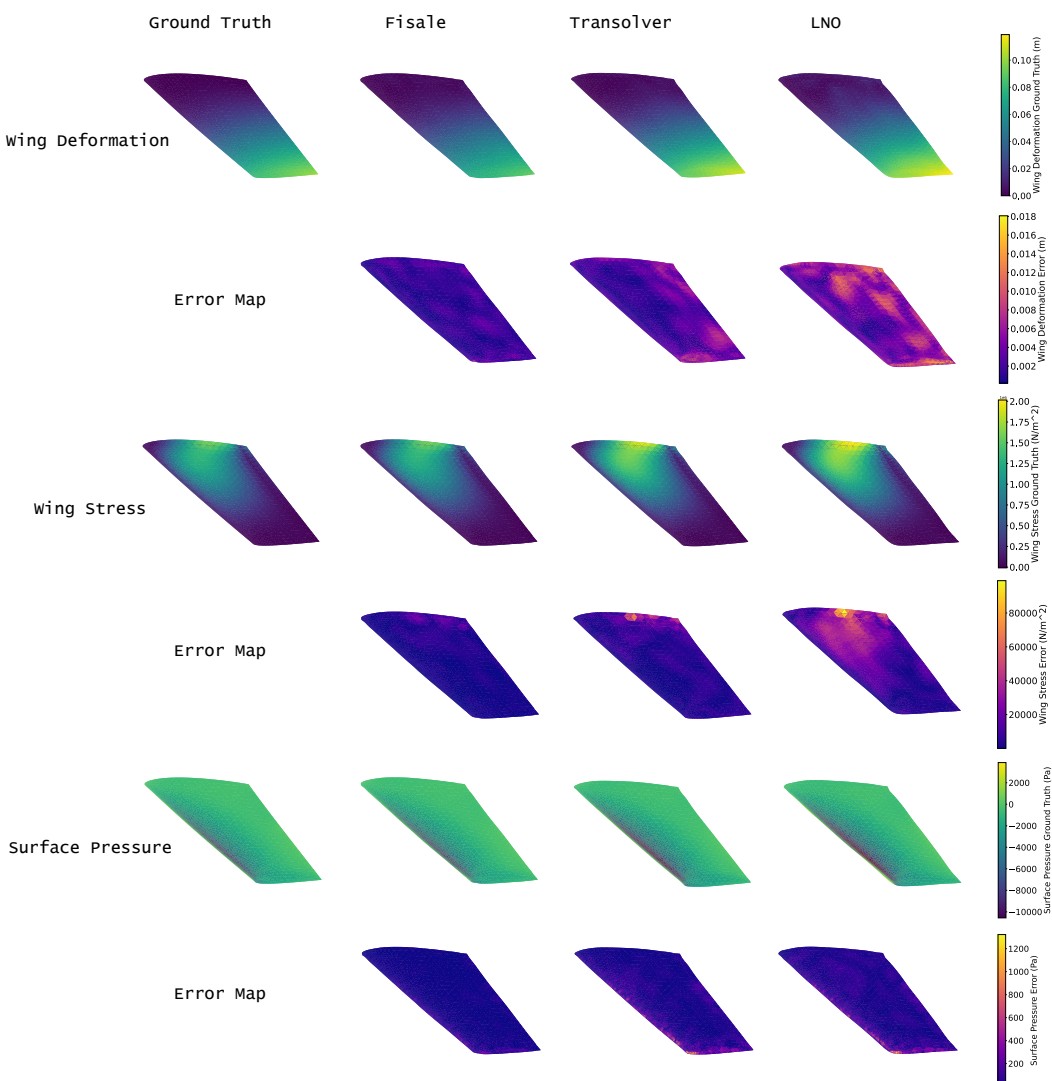

Figure 12: Showcase comparison of venous valve task. The wing deformation, wing stress and surface pressure are shown.

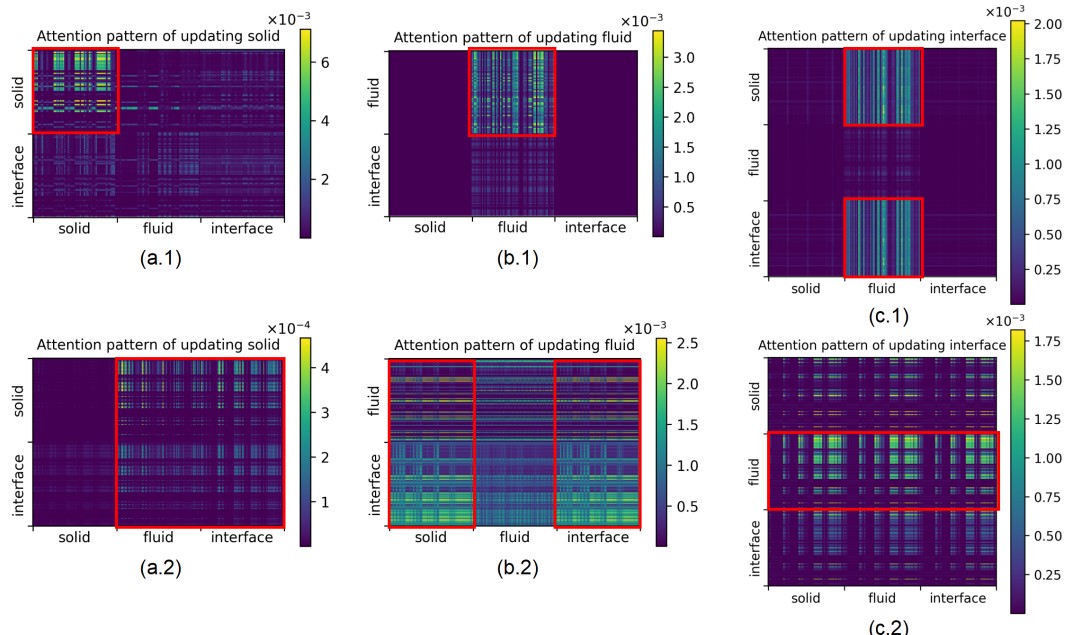

Figure 13: Attention pattern in last layer PCM modules of Fisale. The top row corresponds to the low-resolution latent ALE grid, and the bottom row corresponds to the high-resolution latent ALE grid. Regions with higher activation intensity are highlighted with red boxes. (a.1) and (a.2) represent the updating of the solid component; (b.1) and (b.2) represent the updating of the fluid component; (c.1) and (c.2) represent the interaction on the interface component.

# I    ATTENTION PATTERN

We visualize and analyze the attention patterns in each update step of the PCM module to provide intuitive insight into how attention contributes to modeling the two FSI problem and updating the various physical components. It is worth noting that we use linear attention(Cao, 2021), where the softmax operation is removed and the computation prioritizes $\mathbf{K}^T\mathbf{V}$. Although we cannot directly visualize the attention weights as in standard attention (Vaswani et al., 2017), analyzing $\mathbf{QK}^T$ remains meaningful. On the one hand, from the perspective of matrix multiplication, $\mathbf{Q}(\mathbf{K}^T\mathbf{V})$ and $(\mathbf{QK}^T)\mathbf{V}$ produce the same result. On the other hand, $\mathbf{QK}^T$ explicitly characterizes the correlation between the Query and Key vectors. Even without softmax, the magnitude of its entries reflects the strength with which each Query token aggregates information from the Keys. This form of analysis is also widely used in existing works (Han et al., 2023b; Wu et al., 2024).

Figure 13 illustrates the attention patterns in the last PCM layer of each pathway. The top row corresponds to the low-resolution latent ALE grid, and the bottom row corresponds to the high-resolution latent ALE grid. Regions with higher activation intensity are highlighted with red boxes, indicating where attention tends to aggregate information for the Query.

From the vertical comparison, we can observe two key phenomena:

1. **Complementary patterns across resolutions.** The attention patterns learned on the low- and high-resolution latent ALE grids exhibit clearly complementary behaviors. From the vertical comparison of the highlighted red boxes, we observe that the two resolutions assign different activation strengths to different components. For example, on the low-resolution grid, attention mainly captures the solid-to-solid relationships (Figure 13(a.1)), whereas on the high-resolution grid, it shifts toward capturing interactions between the solid and other components (Figure 13(a.2)). Similar complementary patterns appear across the other attention layers as well. This indicates that the two pathways capture different aspects of the underlying dynamics, further demonstrating the importance of the multi-scale design for modeling two-way FSI.

2. **Different learning tendencies for cross-attention and self-attention across resolutions.** In the cross-attention used for updating the solid and fluid components, the low-resolution grid tends to focus on self-relations (solid-to-solid in Figure 13(a.1) and fluid-to-fluid in Figure 13(b.1)), aggregating information primarily from the same component. In contrast, the high-resolution grid shows the opposite trend, where the solid aggregates information from the fluid and interface (Figure 13(a.2)), and the fluid aggregates information from the solid and interface (Figure 13(b.2)). However, for the interface self-attention, we observe a reversed pattern (Figure 13(c.1) and Figure 13(c.2)): although still complementary across resolutions, both resolutions tend to assign higher activation to inter-region aggregation (e.g., interface-to-solid or interface-to-fluid). Meanwhile, for self-aggregation, the fluid component shows stronger activation (primarily because the fluid region occupies the majority of the computational domain), whereas the solid and interface components exhibit relatively weaker self-focused aggregation. This observation further highlights the importance of the multi-scale PCM. Different components require different aggregation priorities when being updated, and the attention patterns learned on different resolution grids emphasize different regions of influence. By combining these complementary behaviors, the multi-scale PCM ensures comprehensive coverage during updates: capturing both self-related information and cross-component interactions. Therefore, it enables a more complete and physically meaningful representation of complex two-way FSI dynamics.

