# OpenReview forum: "Neural Latent Arbitrary Lagrangian-Eulerian Grids for Fluid-Solid Interaction"
_ICLR.cc/2026/Conference — ICLR 2026 Poster_

### Official Review · Reviewer_k2K6 · 2025-10-27

**Soundness:** 4
**Presentation:** 3
**Contribution:** 4
**Rating:** 8
**Confidence:** 4

**Summary:**

This paper introduces Fisale, a data-driven framework for solving complex two-way FSI problems. Inspired by classical numerical methods such as the Arbitrary Lagrangian-Eulerian (ALE) approach and partitioned coupling algorithms, Fisale excels in three reality-related challenging FSI scenarios, covering 2D, 3D and various tasks by leveraging multiscale latent ALE grids and partitioned coupling module.

**Strengths:**

1. Clear structure and good readability.
2. Innovation in introducing the ALE approach and partitioned coupling module.
3. Abundant experiment and superior experimental results compared with current methods.
4. Rich practical application scenarios.

**Weaknesses:**

1. Lack of consistency in the experimental results: For example, as shown in Figures 8 and 9, the fluid fitting effect of LNO is significantly worse than MGN, but the results in Table 1 are the opposite.
2. Inadequate experiment on complex scenarios: Each experiment in the paper involves relatively low complexity of surfaces and does not prove whether the model has a good fitting ability for more complex surfaces, such as Shape-Net Car (mentioned in AMG).
3. Insufficient proof of generalization ability: The experiment cannot fully demonstrate that the model has learned the physical laws in the fluid dynamics scenario rather than the fitting data in a few scenarios.

**Questions:**

Explain the weaknesses please.

---

> ### Author Response · Authors · 2025-11-21
> **Response 1**
>
> We sincerely appreciate the encouraging overall assessment. We provide our detailed responses to your comments below.
>
>
>
> > **Weakness 1**: Lack of consistency in the experimental results: For example, as shown in Figures 8 and 9, the fluid fitting effect of LNO is significantly worse than MGN, but the results in Table 1 are the opposite.
>
> Thanks for pointing out this. The visualization corresponds to individual sample case, while the quantitative results in Table 1 report statistical averages over the entire test set. It is therefore expected that the ranking of methods on a specific sample may not fully match their ranking in terms of averaged metrics. This mild inconsistency is common in evaluations, especially when comparing models with different inductive biases. Moreover, because our samples include both straight and oscillational beam cases, we observed that LNO performs well in most of the straight scenarios but degrades noticeably in the bending cases. In contrast, MGN behaves more uniformly across both types of samples. This discrepancy explains the difference observed between the visualization results and the averaged metrics.
>
>
>
> > **Weakness 2**: Inadequate experiment on complex scenarios: Each experiment in the paper involves relatively low complexity of surfaces and does not prove whether the model has a good fitting ability for more complex surfaces, such as Shape-Net Car (mentioned in AMG).
>
> We would like to firstly emphasize the motivation behind the scenarios we selected. The **Structure Oscillation** dataset ($\underline{\text{Lines 366–385 of original version}}$), originally proposed in CoDA-NO [1], is a *classical benchmark* for two-way FSI solvers and widely used to validate coupled fluid–solid dynamics. In addition, the **Venous Valve** and **Flexible Wing** experiments represent two real-world FSI scenarios closely related to biomedical engineering and aerodynamics. These settings involve strong two-way coupling, deformable solids, dynamic interfaces, and large geometric changes, making them substantially more challenging and practically meaningful. This is consistent with the goals of AI for Science. Fisale achieves strong performance across all of these tasks, and we will also release the datasets to further support the community.
>
> Regarding the ShapeNet Car task mentioned in AMG, to the best of our knowledge this dataset evaluates **single-phase fluid simulations around rigid bodies**, where the geometry varies but the solid itself does not deform ($\underline{\text{as described in Lines 73-74 of original version}}$). The goal is to predict pressures on the static surface , and many prior works have already demonstrated excellent results on this type of problem. Since our focus ($\underline{\text{Lines 68–76 of original version}}$) is on **two-way FSI**, where **both the fluid evolves and the solid undergoes deformation**, we opted to study tasks that require modeling richer coupled dynamics rather than rigid-body aerodynamic flows. This is more challenging than single-phase medium. We therefore consider our chosen tasks to be better aligned with the intended scope and contributions of this work.
>
> Moreover, we agree that developing more complex and diverse FSI benchmarks would be highly beneficial for the community. We look forward to future work introducing such datasets, and we hope that our contributions, including publicly releasing our own datasets, will help move the field in that direction.

---

> ### Author Response · Authors · 2025-11-21
> **Response 2**
>
> > **Weakness 3**: Insufficient proof of generalization ability: The experiment cannot fully demonstrate that the model has learned the physical laws in the fluid dynamics scenario rather than the fitting data in a few scenarios.
>
> To investigate the model’s generalization ability under different parameter settings in a specific scenario, we conducted a series of OOD experiments ($\underline{\text{Appendix H of original version}}$). In the zero-shot setting, Fisale still outperforms other advanced baselines. The consistent zero-shot performance across unseen physical parameters suggests that Fisale is not merely interpolating training samples but has captured transferable physical patterns.
>
> At the same time, as discussed $\underline{\text{in Appendix K.1 of original version}}$, we acknowledge that task-level generalization, which is a natural proof for generalization ability, remains a significant challenge for data-driven simulators. However, we also want to emphasize the *intended application scenarios* of our method. As described $\underline{\text{in the task setup and in Lines 1810–1821 of original version}}$, Fisale is designed for situations where **fast inference is crucial**, such as repeatedly evaluating the dynamics of a specific class of wings, or scenarios where the **mathematical model is incomplete or not well defined** and only empirical observations are available. In many industrial FSI applications, the cost of a single high-fidelity numerical simulation is extremely high, and traditional solvers cannot take advantage of large amounts of observational data. In such cases, training a data-driven model on task-specific data allows us to **amortize the cost of expensive simulations** and achieve orders-of-magnitude speed-ups at inference time. This task-specific modeling paradigm has also been widely adopted in recent works [1–6] across both fluid and solid mechanics.
>
> Regarding broader generalization, current mainstream approaches rely on training large foundation models across many tasks [7–9]. However, such approaches require substantial amounts of high-quality data. In the fluid domain, many high-quality datasets have been accumulated, so most foundation-model efforts naturally focus on fluid problems. Even so, these works are still largely constrained to Eulerian formulations and regular-domain settings. In contrast, for **two-way FSI**, publicly available high-quality datasets remain scarce compared to single-phase fluids, which limits progress in building foundation-scale FSI models. Nonetheless, we view the construction of multi-task FSI datasets and exploration of task-level generalization as an exciting direction for future work.
>
> ## Reference
>
> [1] Rahman, et al. "Pretraining codomain attention neural operators for solving multiphysics pdes." NeurIPS 2024.
>
> [2] Tao S, et al. "Unisoma: A Unified Transformer-based Solver for Multi-Solid Systems." ICML 2025.
>
> [3] Hao Z, et al. "GNOT: A general neural operator transformer for operator learning." ICML 2023.
>
> [4] Cao, S, et al. "Choose a transformer: Fourier or galerkin." NeurIPS 2021.
>
> [5] Li Z, et al. "Harnessing scale and physics: A multi-graph neural operator framework for pdes on arbitrary geometrie." KDD 2025.
>
> [6] Li, Z, et al. "Fourier neural operator for parametric partial differential equations." ICLR 2021.
>
> [7] Hao Z, et al. "Dpot: Auto-regressive denoising operator transformer for large-scale pde pre-training." ICML 2024.
>
> [8] McCabe M, et al. "Multiple physics pretraining for spatiotemporal surrogate models." NeurIPS 2024.
>
> [9] Herde M, et al. "Poseidon: Efficient foundation models for pdes." NeurIPS 2024.

---

### Official Review · Reviewer_wVhk · 2025-10-27

**Soundness:** 3
**Presentation:** 4
**Contribution:** 3
**Rating:** 6
**Confidence:** 5

**Summary:**

The paper proposes to model fluid-solid interaction (FSI) problems by explicitly modeling fluid/solid/coupling interface as three separate components. The approach boils down to:

1) Projecting state data (positions, features) from all three components onto a shared latent grid
2) Processing data on the grid through a partitioned coupling module (PCM) that sequentially updates solid → grid → fluid → interface
3) Decoding back to the original points

The approach achieves superior performance compared to other methods (GNN and Transformer) that treat the FSI problem as one single unified system.

**Strengths:**

- The paper is well and clearly written, the introduction is comprehensive, and the problem is demonstrated in an intuitive way with Figure 1.
- Architecture design is well-motivated:
  - It suggests a novel approach to handling a gap in prior works, and the problem tackled by the method is practically relevant for many engineering fields.
  - The approach is inspired by established numerical techniques (ALE and partitioned coupling).
  - Each component has a physical justification and is grounded in domain knowledge.
- The method is experimentally validated against multiple baselines and has demonstrated superior performance on multiple tasks. Benchmarks include 2D and 3D problems with different complexity and fidelity.
- The chosen baselines are state-of-the-art methods in physical modelling. The proposed method consistently outperforms all of them.
- Ablation studies are thorough and target specific components of the method (e.g. multiscale ablation and having explicit interface).

**Weaknesses:**

- The paper adopts linear attention which is empirically weaker than standard attention despite the scale of the problems being manageable for flash attention.
- The method does not scale well with increasing the number of latent points, which, in my opinion, is a limitation, see Questions.
- Ablation studies indicate that some of the components might not even be necessary:
  - For example, Table 15 demonstrates that the ordering within PCM doesn't change the performance. Since that is the case, couldn't you simplify the model to:
    1) update solid via cross attention;
    2) update fluid via cross attention;
    3) update interface with self attention;
    4) update grid?

    Note that 1) and 2) can be potentially done in parallel if solid and fluid concatenated, which would make it even faster. Am I missing anything here? Perhaps PCM converges faster?
   - In table 16, substituting the PCM module with attention only slightly drops the performance. However, it is not clear to me what attention is used: linear or self-attention. If not latter, then I would expect much better performance from using standard attention which would, in my opinion, question the value of having the PCM module. It would be great if authors could do the ablation measuring runtime, memory and performance. Having a single kernel might also make it faster, hence I am curious about the runtime.
- The paper does not do an interpretability analysis of the latent ALE grids. While it is not strictly necessary, I do find Transolver slice assignment useful, perhaps the paper would benefit from a similar plot. Besides, studying attention patterns would be a neat addition, similar to how it is done in the EAGLE paper [1].

[1] https://eagle-dataset.github.io/

**Questions:**

I am overall learning towards accept, but answering those question might significantly improve my rating and I also generally improve the paper.
- Did you try using standard attention instead of linear one? The scale appears manageable to me.
- Regarding Table 17, performance vs mesh resolution. It appears to me that the performance does not depend on the mesh resolution. That is counter-intuitive to me as I would expect improved performance with increased number of grid points. Do you have any explanation why it does not happen?
- Which attention is used in Table 16? If linear, would it be possible to update the table with standard attention (flash attention) and compare memory, runtime on top of performance?
- For the set of problems, how does the method compare against a numerical solver?
- Authors report Relative L2 instead of MSE for the CoDA-NO dataset. Is it possible for you to provide MSE just for a direct comparison to the CoDA-NO's paper?
- You claim to be calable to large-scale problems. Do you plan further experiments on large-scale CFD simulations? Would you method be able to handle million point grids?

---

> ### Author Response · Authors · 2025-11-21
> **Response 1**
>
> We sincerely appreciate the encouraging overall assessment. We provide our detailed responses to your comments below.
>
>
> > **Weakness 1**: The paper adopts linear attention which is empirically weaker than standard attention.
>
> > **Weakness 3.2 and Question 3**: What attention is used in table 16: linear or self-attention . It would be great if authors could do the ablation measuring runtime, memory and performance.
>
> > **Question 1**: Did you try using standard attention instead of linear one? The scale appears manageable to me.
>
> These weaknesses and questions are all about the used attentiom mechanism. So we provide a comprehensive response here.
>
> First, we acknowledge that *“linear attention is empirically weaker than standard attention”* can indeed be observed in some AI tasks. However, linear attention is widely used in the AI for PDE community and has shown strong performance on many fluid dynamics benchmarks[1, 2], and we have not observed noticeable disadvantage of linear attention compared with standard attention; in fact, it even performs better in some scenarios. Our motivation comes from the **non-softmax (on attention score) galerkin-type attention proposed in Galerkin Transformer[2], which has been proved to be a kind of neural operator**. This offers theoretical assurance. We also explored the performance of standard attention in our research process. The experiment is conducted on Flexble Wing tasks. The results are as follows:
>
> |  | Relative L2 | Running Time (S) | GPU Memory (GiB) |
> | - | - | - | - |
> | Linear | 0.0136 | 296.30 | 3.10  |
> | Standard | 0.0135 | 301.04 | 3.13 |
>
> We found that, within our design, standard attention **does not offer a noticeable advantage in accuracy, but it does incur a certain drawback in running time**. We believe this is unfavorable for scaling to more complex or larger problems, which constitutes another motivation for our choice: scalibility. We hold the view that **the latent dimension is controllable (i.e, D), whereas the latent grid size (i.e, M) is not**. For some problems, a relatively large latent grid size may be required. However, due to the presence of multi-scale pathways, the latent feature dimension itself does not need to be excessively large. As a result, linear attention can offer clear computational advantages in certain scenarios. This efficiency is particularly beneficial when extending the model to new or larger-scale problems.
>
> Regarding the "simpler module" ablation $\underline{\text{in Table 16 of original version}}$, we would like to clarify that the performance drop is **not slight**. A ~10% degradation represents a **meaningful and practically significant** loss of fidelity. As for the ablation setting, we **did report the result of standard attention**. We also conducted the same ablation on linear attention. The result is listed later. However, we acknowledge that we did not apply FlashAttention for further optimization. To our understanding, FlashAttention is primarily designed to accelerate very large models, whereas our model only has a few million parameters. Therefore, we did not consider applying FlashAttention in original experiments. Nonetheless, we agree that your suggestion is valuable. Therefore, we also experimented with integrating FlashAttention into the ablation study. We directly utilize the function kernel provided in Pytorch: ```sdpa_kernel``` and ```scaled_dot_product_attention```. The results are as follows:
>
> | | Relative L2 | Running Time (S) | GPU Memory (GiB) |
> | - | - | - | - |
> | Fisale | 0.0136 | 296.30 | 3.10 |
> | Simpler Module (Linear Attn) | 0.0146 | 264.23 | 2.35 |
> | Simpler Module (Standard Attn) | 0.0149 | 265.86 | 2.35 |
> | Simpler Module (Flash Attn) | 0.0148 | 264.79 | 2.35 |
>
> Under the same ablation setting, the accuracy, efficiency, and memory consumption of linear and standard attention are very similar. This is partly because attention in our design operates on an $M \times D$ scale. In our setting, when $M$ and $D$ are of comparable magnitude and only a single kernel is used, **the difference between linear and standard attention becomes small**. In addition, the primary memory bottleneck comes from computations on the original sequence of length $N$, so both variants exhibit nearly identical GPU memory usage (which may also be influenced by PyTorch’s memory allocation behavior). Regarding FlashAttention, note that our experiments run on a single RTX 3090 with float32 precision, and the sequence lengths in our design are relatively short. Under these conditions, FlashAttention cannot fully leverage its advantages, which explains why the results are very close to those before optimization. However, we observe that when switching to a single kernel, although the accuracy decreases, there is a substantial improvement in efficiency. We view this as **a meaningful practical optimization: in scenarios where efficiency is more critical than accuracy**, this configuration can be a favorable choice.

---

> ### Author Response · Authors · 2025-11-21
> **Response 2**
>
> In summary, we observe that linear attention provide comparable performance over standard attention due to the sequence scale. In the PCM, the key factor is the **iterative partitioned coupling strategy**, while attention simply serves as a mechanism for cross-domain interaction. This interaction can take many forms and is **not fundamentally determines model performance**. In contrast, linear attention offers a theoretically grounded [2] and computationally scalable formulation, which makes it well suited for extending the model to more complex problems. This is also why linear attention has been widely adopted in many AI for PDE works. Nonetheless, we appreciate you for pointing this out, as it has prompted us to think more deeply about potential directions for improving both performance and efficiency.
>
> > **Weakness 2 & Question 2**: The method does not scale well with increasing latent points.
>
> We agree that, in classical numerical solvers, increasing mesh resolution typically improves accuracy. However, in our framework the latent ALE grid plays a **very different role** from a physical discretization of the PDE.
>
> Firstly, the latent ALE grid is *not* used to discretize the governing equations. Instead, it serves as a **geometry-aware latent representation** that aggregates information from the physical particles/mesh through a continuous, distance-based interpolation ($\underline{\text{Lines 266–231 of original version}}$). Because this interpolation step maps all physical information onto the latent grid regardless of its resolution, the latent grid **does not lose information when the number of grid points changes**; it simply provides a different set of sampling centers over the same continuous field. This means that the latent representation behaves more like a *smooth functional embedding*, a smooth latent field from which the grid nodes act as sample points, than a numerical discretization. Therefore, the model is inherently **less sensitive to the specific latent grid resolution** than input physical discretization.
>
> Secondly, Table 17 uses a single-scale latent pathway (H = 1). In this setting, increasing grid resolution does not significantly increase expressive power, because the representation is structurally limited to a single scale. As we claimed $\underline{\text{in Lines 1455-1460 of original version}}$, this means the model cannot simultaneously represent fine-grained local behavior and large-scale global dynamics. Its **“expressive bandwidth” is structurally capped**. This is also aligned with our observation from $\underline{\text{the attention pattern in the revised version}}$. As a result, increasing grid resolution within a single-scale architecture adds redundancy rather than new modeling capability. This is consistent with $\underline{\text{Table 18 and Fig. 7 of original version}}$, where adding *multi-scale* pathways (H = 2) yields a much larger performance improvement than increasing the single-scale grid resolution.
>
> > **Weakness 3.1**: Note that 1) and 2) can be potentially done in parallel if solid and fluid concatenated, which would make it even faster.
>
> As we have claimed $\underline{\text{in Lines 1416-1421 of original version}}$, the PCM is inherently a flexible and robust framework.  Fisale adopts a stacked architecture, the update order of components does not need to remain fixed within each layer. Instead, the interactions among fluid, solid, grid, and interface can be iteratively adjusted through vertical information flow across layers. This iterative propagation helps compensate for local order choices, allowing the model to refine cross-domain interactions over multiple stages. Our choice is a **physically reasonable ordering**.
>
> As for your suggestion about parallel implementation, from the physical insight (as described $\underline{\text{in Lines 156-178 of original version}}$), the change of fluid or solid influences the other one. This should be a sequential and interdependent operation. However, from a practical implementation perspective, the parallel operation is indeed valuable, as parallelizing these updates could eliminate one attention operation. We experiment with this idea and obtained the following results.
>
> | | Relative L2 | Running Time (S) | GPU Memory (GiB) |
> | - | - | - | - |
> | Ours | 0.0136 | 296.30 | 3.10 |
> | Parallel | 0.0135 | 295.11  | 3.15  |
>
> Although this modification removes one attention operation, our experiments indicate that this part is not the main computational bottleneck of Fisale because of $M\ll N$, nor does it lead to noticeable improvements in model convergence. The slightly increased GPU memory may because the additional intermediate tensors of both domains at the same time. This is mainly caused by the PyTorch’s memory caching and allocation strategy. We really appreciate the suggestion, and we will continue to deeply think whether certain computation steps can be merged or reorganized to further improve performance.

---

> ### Author Response · Authors · 2025-11-21
> **Response 3**
>
> > **Weakness 4**: The paper does not do an interpretability analysis of the latent ALE grids.
>
> Thank you for the insightful suggestion. We agree that interpretability analyses could further enhance the clarity of our model’s behavior. For better understanding the interactions among components, we visualize the attention patterns within the PCM module. **We have added the visualization and analysis into the revised version**. Because we cannot add figures in the rebuttal response, please see $\underline{\text{ Appendix K in revised version}}$ for detailed content. We only provide two key phenomena here:
>
> 1. **Complementary patterns across resolutions.**
>
> 2. **Different learning tendencies for cross attention and self attention across resolutions.**
>
> > **Question 4**: For the set of problems, how does the method compare against a numerical solver?
>
> $\underline{\text{As noted in Appendix B of original version}}$, due to the lack of real-world FSI measurements, our datasets are generated using COMSOL. Using numerical solvers to generate dataset is also a common practice in many AI for PDE works [3-5]. Under this setting, it is unrealistic for a data-driven model to *outperform* the numerical solver that produced the ground-truth data. However, our goal is fundamentally different: we aim to **significantly improve computational efficiency while keeping the approximation error as small as possible**.
>
> For example, in the Flexible Wing task, simulating a single COMSOL sample requires **approximately two hours** of computation. Moreover, this computation cannot be reused when solving the same task under different parameter settings. The numerical solver must run from scratch each time. This highlights a limitation of classical solvers. In contrast, our model, once trained, can generate predictions **within milliseconds**, providing orders-of-magnitude speed-up while retaining high accuracy. This efficiency gain is precisely the motivation behind AI for PDEs and neural operator methods.
>
> Looking ahead, we also note that numerical solvers themselves have modeling assumptions and numerical errors that introduce a gap between simulation and real-world behavior. In practical scenarios where real measurements become available, data-driven models can directly learn from such data, potentially achieving solutions that are **both faster and more accurate** than traditional solvers. This is **an important long-term goal for our line of research**.
>
>
>
> > **Question 5**: Is it possible for you to provide MSE just for a direct comparison to the CoDA-NO's paper?
>
> Firstly, let me briefly summarize the Structure Oscillation setup described in the CoDA-NO paper. CoDA-NO is a pretrained model whose primary focus is **few-shot generalization**, as demonstrated in its Table 1 and settings. This is directly comparable to the ood evaluation we performed in Table 19 of our paper, where the model is tested on data with $\mathrm{Re} = 4000$. The evaluation like we did in Table 1 of our paper is not found in their paper. Therefore, we convert our ood results into MSE format to enable a direct comparison with Table 1 of CoDA-NO. The results are as follows:
>
> | |0| 5| 25| 100|
> |-|-|-|-|-|
> | CoDA-NO | - | 0.308 | 0.143 | 0.069 |
> | Fisale | 0.131 | - | - | - |
>
> It is also worth noting that our OOD results are obtained **without any fine-tuning**, whereas Table 1 in CoDA-NO reports performance **after fine-tuning on samples**. Although this is not perfectly aligned, the comparison still provides a meaningful reference: even without fine-tuning, our model performs better than CoDA-NO with fine-tuning on 5 or 25 samples. This can provide a direct and intuitive comparison with CoDA-NO.

---

> ### Author Response · Authors · 2025-11-21
> **Response 4**
>
> > **Question 6**: Do you plan further experiments on large-scale CFD simulations? Would you method be able to handle million point grids?
>
> $\underline{\text{As shown in Table 13 of original version}}$, Fisale is significantly more GPU-memory friendly than existing attention-based simulators. For example, it requires only about **3 GB of GPU memory for >35,000 mesh points**, which suggests strong potential for scaling toward much larger problem sizes. Given the linear-complexity design of our architecture, we believe that extending Fisale to million-point grids is feasible in principle.
>
> However, conducting experiments at such scale requires suitable datasets. In the pure fluid setting, large-scale CFD benchmarks are becoming increasingly available. For instance, Transolver++[6] demonstrates excellent results on high-resolution Eulerian flows. However, our focus is **two-way FSI**, which involves both fluid evolution and solid deformation. For this domain, **large-scale, high-fidelity datasets simply do not yet exist**. This scarcity is currently a major challenge for the community and limits systematic investigation of million-scale FSI simulations.
>
> We view this as an important direction for future work. The datasets used in our paper contribute toward filling this gap, and we will make them publicly available to help support the development of larger and more complex benchmarks. We hope that releasing these datasets will encourage further research on scalable models and ultimately push the field toward truly large-scale two-way FSI simulations.
>
> ## Lastly
>
> Lastly, we would like to thank you for your constructive comments that aim to improve our work. We greatly appreciate the opportunity to discuss potential improvements and future extensions of our work, especially about the motivation, analysis, goal and future. Such discussions are highly valuable for generating new ideas and shaping the direction of the field. They have also prompted us to think more deeply about our own approach and paradigm. In many ways, these reflections are even more important to us than the paper itself. Thank you once again, sincerely.
>
> ## Reference
>
> [1] Hao Z, et al. "GNOT: A general neural operator transformer for operator learning." ICML 2023.
>
> [2] Cao, S, et al. "Choose a transformer: Fourier or galerkin." NeurIPS 2021.
>
> [3] Li, Z, et al. "Fourier neural operator for parametric partial differential equations." ICLR 2021.
>
> [4] Tao S, et al. "Unisoma: A Unified Transformer-based Solver for Multi-Solid Systems." ICML 2025.
>
> [5] Pfaff, et al. "Learning Mesh-Based Simulation with Graph Networks." ICLR 2021.
>
> [6] Luo H, et al. "Transolver++: An Accurate Neural Solver for PDEs on Million-Scale Geometries." ICML 2025.

---

> > ### Comment · Reviewer_wVhk · 2025-11-26
> > **Response by Reviewer wVhk**
> >
> > I am grateful to the authors for their effort and the comprehensive response which adequately addresses my concerns (attention benchmark, interpretability analysis, method structure ablation). I furthermore raise my score to Accept (8) as I believe the submission now surpasses the quality bar of ICLR.

---

> > > ### Author Response · Authors · 2025-11-26
> > >
> > > Thank you very much for your recognition. Your suggestions have greatly improved the quality of our work. We will continue refining the paper based on the remaining reviewers’ comments. We sincerely appreciate your support, once again.

---

### Official Review · Reviewer_yXgU · 2025-10-31

**Soundness:** 1
**Presentation:** 3
**Contribution:** 2
**Rating:** 2
**Confidence:** 3

**Summary:**

This work introduces Fisale, a purely data-driven surrogate model for two-way fluid structure interaction between incompressible fluids and deformable solids. The combined solid-fluid + interface state is represented using multiscale latent ALE grids and the dynamic interface is evolved in time using self and cross attention mechanisms in a partitioned interative fashion. Multiscale modeling is handled using parallel processing of latent ALE grids, interspersed with aggregation after every layer, to capture phenomena at various scales. The authors test the effectiveness of their method by predicting future states in three different scenarios: Structure Oscillation, Venous Valve and Flexible Wing, and compare the obtained frames with the numerically simulated ground truth as well as other neural models.

**Strengths:**

1.	Multiscale modeling has proved to be an effective way in solving PDEs which involve dynamic boundaries and multiple domains, such as in fluid-structure interaction. Using more samples in interface regions for accuracy while using less samples is domain interiors for efficiency is a well-studied approach. The authors seem to leverage this well in the construction of their architecture.
	2.	In the experiments provided in the paper, Fisale seems to outperform other baselines, which is a good indication.
	3.	The experimentation in the main paper, together with the Appendix, is quite extensive, with a great number of relevant baselines.

**Weaknesses:**

1.	Although the authors claim to leverage the classical numerical formulations like ALE and Partitioned Coupling, it does not seem to be the defining factor here. This is clear from the ablations performed in Appendix G. The increased accuracy in the experiments in the main paper can simply arise from the high representational capacity of the architecture itself, since there are a lot of learnable components. Also, the choice of the dimension of the latent ALE grid (i.e. D) does not seem to be explicitly defined anywhere in the paper. A large value of D can also cause overfitting.
	2.	The intuition behind using attention for coupling is not very clear. In my understanding, attention mechanisms are used to model long-range dependencies which arise in text based domains, for example. The interface dynamics in fluid-structure interaction are necessarily short range, as far as both fluid and solid domains are concerned. So, using attention would be wasteful, and might even be numerically unstable.
	3.	The proposed model does not seem to be generalizable across tasks, which involve different geometry (static and dynamic). There is separate training and dataset for each of the three tasks. In that case the gain in inference time is not so much to offset the training time required for each of the tasks, when compared with a classical numerical simulation.
	4.	There is no mention of how the initial values of physical quantities q is fed to the model. The only point where it seems to be involved is in defining the initial latent state x^{0,h} = Linear(u). Even that uses a trainable linear layer. Without information about the initial state, no model can accurately predict future states. This supports my earlier observation that the proposed model may just be overfitting the data and not actually learning the dynamics.
	5.	 The limitations should be mentioned in the main paper, not in the Appendix.

**Questions:**

1.	The authors should consider expanding some of the captions for exposition purposes, especially for Figure 2 and Table 3
	2.	In this work, I believe that even the grid sample positions are inferred from the model, along with the sample values, which is good. But how are the sample positions defined for the initial state. Specifically, how is g defined for the initial state?

---

> ### Author Response · Authors · 2025-11-21
> **Response 1**
>
> We sincerely thank you for the insightful and constructive comments. We provide our detailed responses to your comments below.
>
> > **Weakness 1**: Although the authors claim to leverage the classical numerical formulations like ALE and Partitioned Coupling, it does not seem to be the defining factor here. Also, the choice of the dimension of the latent ALE grid (i.e. D) does not seem to be explicitly defined anywhere in the paper.
>
> The numerical formulations we draw from, namely the ALE motion and the partitioned coupling strategy, directly shape **how the network processes information**.As shown in  $\underline{\text{Table 16 of original version}}$, replacing the PCM with a single attention block (“Simpler Module” ablation) leads to a **clear degradation (~10%)**. Within the ablation setting, we adjust the latent dimension to keep their parameter count approximately the same, ensuring comparable capacity. This demonstrates that the improved performance does not simply arise from representational capacity, but from the *structured inductive biases* embedded by ALE-based grid motion, partitioned substeps, and interface-aware decomposition. These priors help the model **better capture cross-domain dependencies** that arise in two-way FSI.
>
> Moreover, we have carefully **controlled model size when comparing against baselines** ($\underline{\text{Table 9 of original version}}$), ensuring that Fisale uses a comparable or even smaller number of parameters than several strong baselines. Moreover, in the ablation where we increase grid resolution or add more pathways ($\underline{\text{Table 18 and Figure 7 of original version}}$), we observe diminishing returns rather than unbounded gains, which further suggests that capacity alone is not the determining factor. The structured numerical inductive biases are the primary contributors to stable learning and generalization.
>
> As for the choice of the dimension of the latent ALE grid (i.e., D), we have provided **detailed model configuration for each task in $\underline{\text{Table 8 of original version}}$** of original submission version.
>
> > **Weakness 2**: The intuition behind using attention for coupling is not very clear.
>
> $\underline{\text{As we have claimed in Lines 289-292 of original version}}$, **attention mechanisms are well-suited for modeling PDE-related physical systems** due to their strong capacity to capture nonlinear dependencies, long-range interactions, and perform spatial aggregation across irregular domains [1-3]. Recent studies have further shown that attention itself can be interpreted as an integral operator, capable of approximating complex mappings across function spaces [1, 4, 5]. Moreover, the flattened sequence of the latent ALE grid aligns naturally with the input format required by attention mechanisms, making attention a seamless and effective choice for updating physical states across fluid, solid, and interface domains. We have added more intuition and insight of using attention into the revised version for clarification.
>
> We would like to further clarify that FSI is **not purely short-range**: it contains both short-range, autoregressive dependencies (e.g., local interface traction and mesh motion) and long-range, state-to-state dependencies (e.g., global pressure redistribution, large-scale solid deformation, and steady-state convergence). Our design therefore deliberately combines two mechanisms: **Laplacian-based ALE mesh-velocity updates** to robustly capture short-range geometric and interface interactions, and **attention-based coupling** to capture global, long-range effects that cannot be resolved by purely local operators. Moreover, many recent works [1-3] also show that attention can effectively model both short- and long-range physical interactions.
>
> Regarding concerns of wastefulness or numerical instability, our analysis and experiment in $\underline{\text{Appendix F of original version}}$ show that the computational complexity of Fisale remains **linear in the scale of the input problem $N$**, due to the structure of the latent ALE grids. $\underline{\text{Table 13 of original version}}$ further demonstrates that the model is highly GPU-efficient and able to handle large-scale problems (e.g., >35k points) without incurring the quadratic cost typically associated with dense attention. As for numerical stability, we **have not observed instability** in any of our experiments; Fisale consistently produces stable rollouts across multiple seeds and settings, and we have likewise not observed instability in other attention-based physics models [1, 3, 4]. For these reasons, we believe that attention is a well-justified and effective choice for coupling in our architecture.

---

> ### Author Response · Authors · 2025-11-21
> **Response 2**
>
> > **Weakness 3**: The proposed model does not seem to be generalizable across tasks.
>
> As we discussed $\underline{\text{in Appendix K.1 of original version}}$, task-level generalization is a real challenge for purely data-driven models, especially when the parameter budget is small (Fisale has <10M parameters). We acknowledge this limitation. However, we also want to emphasize the *intended application scenarios* of our method. As described $\underline{\text{in the task setup and in Lines 1810–1821 of original version}}$, Fisale targets settings where **fast inference is crucial**, such as repeatedly evaluating the dynamics of a particular class of wings, or scenarios where the **mathematical model is incomplete or not well defined** and only empirical observations are available.
>
> In many industrial applications, especially in FSI, the cost of a single high-fidelity numerical simulation is extremely high, and traditional solvers cannot leverage large amounts of observational data. In these cases, training a data-driven model on task-specific data allows us to **amortize the cost of expensive simulations** and achieve orders-of-magnitude speed-ups during inference, which directly benefits design optimization, iterative prototyping, or uncertainty exploration. This modeling paradigm, training a task-specific data-driven simulator, has also been adopted by many recent works [1-6] in both fluid and solid mechanics.
>
> We also agree that **multi-task generalization is an important direction**. The current mainstream approach is to train larger foundation models across many tasks [7-9]. However, such approaches require substantial amounts of high-quality data. In the fluid domain, many high-quality datasets have been accumulated, so most foundation efforts naturally focus on fluid problems. Even so, these works are still largely limited to Eulerian formulations and regular-domain fluid settings. For two-way FSI, however, publicly available high-quality datasets are significantly more limited compared with single-phase fluids, which constrains such efforts at present. Nonetheless, we view building multi-task FSI datasets and exploring task-level generalization as an exciting direction for future work. The datasets used in our work contribute toward filling this gap, and we plan to release them publicly to support the development of multi-task and foundation models for two-way FSI in the future.
>
> > **Weakness 4**: There is no mention of how the initial values of physical quantities q is fed to the model.
>
> We would like to clarify that the initial physical quantities **q are indeed explicitly provided to the model**. $\underline{\text{As defined in Lines 182–187 of original version}}$, each point's input feature $u$ is composed of its geometry $g$ together with its physical state $q$ (fluid: $\mathbf{u}_f = \text{Concat}[\mathbf{g}_f \in \mathbb{R}^{N_f \times d}, \mathbf{q}_f \in \mathbb{R}^{N_f \times C_f}]$, solid: $\mathbf{u}_s = \text{Concat}[\mathbf{g}_s \in \mathbb{R}^{N_s \times d}, \mathbf{q}_s \in \mathbb{R}^{N_s \times C_s}]$ and interface: $\mathbf{u}_b = \text{Concat}[\mathbf{g}_b \in \mathbb{R}^{N_b \times d}, \mathbf{q}_b \in \mathbb{R}^{N_b \times C_b}]$). **They are concatenated as a complete representation for each point**. We have clarified this $\underline{\text{in Section 3 of revised version}}$. This is a standard way of feeding points' information into neural simulators. Furthermore, as described $\underline{\text{in Lines 918–941 of original version}}$, the network receives the features $u^t$ at the **current timestep $t$**, which corresponds precisely to the “initial values” referenced in your comment, when initializing the latent representations $x^{0,h}$. The linear layer applied at initialization is simply a learnable feature encoder, not a mechanism that removes or ignores the input physics. Thus, the model always conditions on the full physical state at the beginning of each rollout segment. Given that the initial state is explicitly incorporated and used throughout the update steps, the observed accuracy cannot be attributed to overfitting due to missing physical inputs; instead, it reflects the model’s ability to learn the coupled dynamics from the data.
>
> > **Weakness 5**: The limitations should be mentioned in the main paper, not in the Appendix.
>
> > **Question 1**: The authors should consider expanding some of the captions for exposition purposes, especially for Figure 2 and Table 3.
>
> Thank you for the constructive suggestions. Due to space constraints in the original submission, we placed the limitations section in the appendix. We have moved the limitations into the main text in the revised version ($\underline{\text{Section 5 of revised version}}$). Moreover, we have also added some captions to improve the clarification ($\underline{\text{Figure 2 and Table 3 of revised version}}$). Please see the revised version for revisions.

---

> ### Author Response · Authors · 2025-11-21
> **Response 3**
>
> > **Question 2**: In this work, I believe that even the grid sample positions are inferred from the model, along with the sample values, which is good. But how are the sample positions defined for the initial state. Specifically, how is g defined for the initial state?
>
> If I understood correctly, you are referring to how the *initial sample positions* $g$ (i.e., the initial mesh coordinates) are defined. If my understanding of your concern is incorrect, please feel free to clarify.
>
> $\underline{\text{As stated in Lines 182–187 of original version}}$, $g$ represents the spatial coordinates of the discretization points of the physical domain. In practice, the initial mesh for each dataset is generated using standard **finite element mesh generation techniques**, which is a conventional way to discretize a continuous domain in classical numerical solvers. These meshes are fully determined by the geometry of each problem setup (e.g., the cylinder–beam configuration, venous valve geometry, or flexible wing shape), and are not learned by the model.
>
> During inference, the model takes these initial coordinates, together with their associated physical quantities $q$, as the input $u$. The latent ALE grids are then initialized and updated on top of this fixed physical mesh according to our design. Thus, the model does not infer the initial mesh positions; rather, it receives physically meaningful initial coordinates and quantities from the FEM-generated mesh as its input.
>
> ## Reference
>
> [1] Wu H, et al. "Transolver: A fast transformer solver for pdes on general geometries." ICML 2024.
>
> [2] Tao S, et al. "Unisoma: A Unified Transformer-based Solver for Multi-Solid Systems." ICML 2025.
>
> [3] Hao Z, et al. "GNOT: A general neural operator transformer for operator learning." ICML 2023.
>
> [4] Cao, S, et al. "Choose a transformer: Fourier or galerkin." NeurIPS 2021.
>
> [5] Li Z, et al. "Harnessing scale and physics: A multi-graph neural operator framework for pdes on arbitrary geometrie." KDD 2025.
>
> [6] Li, Z, et al. "Fourier neural operator for parametric partial differential equations." ICLR 2021.
>
> [7] Hao Z, et al. "Dpot: Auto-regressive denoising operator transformer for large-scale pde pre-training." ICML 2024.
>
> [8] McCabe M, et al. "Multiple physics pretraining for spatiotemporal surrogate models." NeurIPS 2024.
>
> [9] Herde M, et al. "Poseidon: Efficient foundation models for pdes." NeurIPS 2024.

---

> ### Author Response · Authors · 2025-11-27
>
> Dear Reviewer,
>
> We hope this message finds you well. As the discussion phase is approaching its conclusion in less than a week, we would like to kindly ask whether our previous responses have addressed your concerns, and whether you have any further questions or suggestions you would like us to discuss. We truly appreciate this valuable opportunity for discussion. Your feedback is extremely valuable for improving the quality of our work. Thank you very much for the time and effort you have dedicated to our submission.
>
> Sincerely,
>
> Authors

---

### Official Review · Reviewer_yad9 · 2025-10-31

**Soundness:** 2
**Presentation:** 1
**Contribution:** 2
**Rating:** 4
**Confidence:** 3

**Summary:**

The authors propose a new neural operator framework for handling fluid-solid interaction problems with cross-domain awareness. The proposed architecture Fisale leverages multi-scale latent grids to provide geometry aware embeddings. They further enable problem decomposition to handle nonlinear dependencies.

**Strengths:**

1. The paper provides good motivation to model ALE systems for fluid-solid interactions.
2. The authors conduct thorough experiments, with a very detailed appendix section.
3. The authors have considered SOTA baselines and SOTA problem settings.

**Weaknesses:**

1. The writing is quite dense in section 3 and it is not clear to me exactly how this proposed architecture compares to "A Neural Material Point Method for Particle-based Emulation, O Sharabi"
2. It seems like grid update is similar to message passing. It seems like section 3.3 is describing typical neural network operations and as such can be moved to the appendix for better readability.
3. The overall writing can be significantly improved, with many parts being unclear with substantial focus on describing techniques that are well established in ML community (such as update fluid state and update interface influence) which describe the sequence of operations performed within the network, without any intuition for the said operations. Similarly, "update grid coordinate" is overly detailed.

**Questions:**

1. It was shown in GNS that the model is able to handle multi-material systems (water-sand, water-jelly) by simply creating material-type embedding and leveraging data-driven training to learn the interaction dynamics. How does that compare against the proposed approach?
2. Lagrangian systems typically require GNN based processing and models designed to handle Eulerian grid inputs fail to handle inter-particle interactions. This was shown in UPT, Transolver and GIOROM. The problem setting describes a system that is Eulerian or Lagrangian but the results don't seem to include any interaction dynamics datasets despite referencing GNS several times in the paper.
3. The rationale behind KNN based grid is not fully clear to me. Typically neural operators leverage radius based grids with mean aggregation to enable discretization invariance (GINO, multipole graph kernel network). This is because when the input discretization changes, radius aggregation ensures an entire region is captured. With KNN based grids, the aggregation region changes based on discretization -- how does this enforce discretization convergence?
4. The approach inherently handles multi-scale inputs. It would be interesting to see if it also handles AMR, as part of future work.

---

> ### Author Response · Authors · 2025-11-21
> **Response 1**
>
> We sincerely thank you for the insightful and constructive comments. We provide our detailed responses to your comments below.
>
> > **Weakness 1.2**: Difference between Fisale and  "A Neural Material Point Method for Particle-based Emulation, O Sharabi"
>
> We firstly response the second part of weakness 1. We appreciate the opportunity to clarify the conceptual differences between our framework and NeuralMPM ("A Neural Material Point Method for Particle-based Emulation, O Sharabi").  We summarize the key distinctions below.
>
> - NeuralMPM uses a **fixed Eulerian voxel grid** purely for particle-to-grid projection and grid-to-particle back-projection. All physical updates occur on this grid using image-to-image networks. Fisale, inspired by ALE methods, constructs latent ALE grids that (1) deform according to the learned latent grid velocity; (2) carry unified multi-physics embeddings; (3) serve as cross-domain coupling hubs. Our grid is not a voxelization of particles but a **learnable and deformable latent representation** that bridges Lagrangian/Eulerian regions and evolves with the interface.
>
> - NeuralMPM performs a **monolithic grid projection and update**, treating the entire domain uniformly. In contrast, Fisale **treats the interface as an explicit module** with its own dedicated embeddings and introduces a **Partitioned Coupling Module (PCM)** that mirrors classical partitioned FSI solvers to decompose the update process in an **iterative, domain-aware** fashion. This design enables the model to capture interface coupling dynamics that are critical in two-way FSI. As further evidenced by the ablation results $\underline{\text{in Table 14 and 16 of original version}}$, this cross-domain update paradigm provides the ability to disentangle and coordinate domain-specific interactions. These capabilities are essential for two-way FSI systems and are absent in NeuralMPM’s monolithic formulation.
>
> - NeuralMPM is fundamentally built around explicit time-stepping: its dynamics are defined as a strictly autoregressive particle–grid update that advances the system by one (or a small bundle of) timesteps at a time. The **local receptive field** of the processor captures only short-range interactions. In contrast, Fisale introduces **global latent representations**, and uses a partitioned attention-based update scheme  which allows multi-step reasoning within a single forward pass. This design breaks the dependence on step-by-step temporal integration and reduces the locality constraints of the processor, enabling our model to capture long-range behavior and infer stable or steady states.
>
> > **Weakness 1.1**: The writing is quite dense in section 3.
>
> > **Weakness 2**: It seems like grid update is similar to message passing and section 3.3 is describing typical network operations.
>
> > **Weakness 3**: The overall writing can be significantly improved, with many parts describing well-established operations within the network, without any intuition for the said operations. "update grid coordinate" is overly detailed.
>
> The weaknesses 1.1, 2, and 3 are all related to writing clarity, and we provide a consolidated response below.
>
> Firstly, our writing strategy is to present the model in a **top-down manner**, starting from the overall architecture and then detailing each module. We intentionally provide a **step-by-step explanation of the computation pipeline** to ensure transparency and reproducibility of the model. This style follows common practice in neural simulator literature [1–3], where the algorithmic flow and update rules are explicitly presented to avoid ambiguity during reading and implementation.
>
> Secondly, instead of only presenting the order of operations ,we also provide detailed **intuition and insight behind each update step**. We believe that in AI for Science, beyond proposing entirely new “fancy’’ architectures, it is equally important to explain **why existing ML components are appropriate and how they encode physical intuition**. $\underline{\text{As stated in Lines 156–179 and 286–288 of original version}}$, the PCM module explicitly **follows the classical partitioned coupling algorithm**, updating the solid, grid, fluid, and interface in a physically meaningful sequence. Our exposition mirrors this solver-inspired structure. For instance, $\underline{\text{Lines 289-293 of original version}}$ describe why attention is suitable for updating each domain, how position embeddings are used, and how the $Q$, $K$ and $V$ formulation reflects the underlying physics. Similarly, $\underline{\text{Lines 304–319 of original version}}$ explain the motivation behind the ALE-based grid update and how it corresponds to the ALE mesh-velocity formulation rather than generic neural processing. Except for Section 3.3, we also have many descriptions related to our design insight and intuition in $\underline{\text{Section 3.1 and 3.2 of original version}}$ about the latent ALE grid and physcial encoding and decoding.

---

> ### Author Response · Authors · 2025-11-21
> **Response 2**
>
> Thirdly, Section 3.3 does not aim to describe standard operations. Its purpose is to explain **how each physical component is updated within each iteration**, following the idea of a partitioned coupling algorithm, and to make the design rationale behind these updates explicit. As for the grid update, it is derived from the ALE principle that **mesh velocity satisfies a diffusion-like balance** ($\underline{\text{Section 2 and Lines 305 of original version}}$). Although the discretization form can appear superficially similar to message passing ($\underline{\text{as acknowledged in Lines 313–315 of original version}}$), this is merely a **mathematical coincidence** arising from discretizing the ALE PDE, not from directly adopting a GNN-style paradigm. We describe this process concretely to **highlight the underlying ALE insight** that motivates our grid update design.
>
> Nonetheless, we fully acknowledge your concern that some parts may not be optimized for a broader readership. We have revised the manuscript to make the intuition explicit and remove some unnecessary details. Concretely, (1) we remove and trim some detailed descriptions in Section 3.3 like the update of grid and attention operation; (2) we add more intuition like why using attention. Please see $\underline{\text{Section 3.3 of revised version for revisions}}$. We sincerely appreciate your suggestion, which will help us strengthen the clarity.
>
> > **Question 1**: Compare with GNS.
>
> GNS [4] is indeed a highly influential work in AI-based simulation. Its successor MGN [5] is from the same research group and adopts the same GNN-based message-passing architecture. They are different from the construction of the input graph.  As a result, **comparing with MGN is effectively equivalent to comparing with GNS** in terms of modeling capability. In our experiments, we include MGN as well as its more recent improved variants, HOOD and AMG. $\underline{\text{As shown in Tables 1, 2, and 3}}$, our method consistently outperforms all of these GNN-based simulators across tasks and domains. Moreover, scalability is another important distinction. In the Flexible Wing task we use meshes with over 35,000 points. $\underline{\text{As shown in Table 13 of original version}}$, such large-scale settings pose significant computational challenges for GNN-based simulators due to their graph construction and message-passing cost. In contrast, our architecture maintains **linear complexity with respect to the number of physical points**, enabling efficient and stable learning in large 3D FSI scenarios. This scalability advantage is essential for handling realistic engineering geometries. Overall, while GNS/MGN-style methods are strong baselines for general particle or mesh-based simulation, our approach is designed specifically for two-way FSI and provides both architectural advantages and improved scalability.
>
> > **Question 2**: Lagrangian systems require GNN processing and models designed to handle Eulerian grid fail to handle inter-particle interactions. The problem setting describes a system that is Eulerian or Lagrangian but the results don't include any interaction dynamics datasets.
>
> While several prior works emphasize that Lagrangian systems are often handled using GNN-based processors, this does not imply that Transformer-based architectures cannot model Lagrangian dynamics. In fact, many studies [2, 6] have demonstrated that **Transformers can naturally operate on unordered point sets**, treating particles or mesh nodes as a sequence.
>
> Regarding the concern about “interaction dynamics datasets,” we would like to clarify that our experiments do include rich interaction dynamics. The **Structure Oscillation** dataset used in our first experiment, originally proposed in CoDA-NO [7], is a *classical two-way FSI benchmark* widely used for validating FSI solvers (see $\underline{\text{Lines 366–385 of original version}}$). It involves **strong bidirectional coupling between a deformable solid and surrounding incompressible fluid**, and is significantly more complex than the one-way settings often studied in prior ML simulation work.
>
> In addition, our **Venous Valve** and **Flexible Wing** experiments represent two real-world, highly coupled multiphysics systems from biomedical engineering and aerodynamics. Compared with interaction datasets commonly used in GNS, such as cube drop or water drop, our tasks feature **large deformations, dynamic interfaces, and complex geometry-driven coupling**, making them more challenging. This is more aligned with the motivations of AI for Science that focusing on real-world problems. Fisale achieves strong performance on all these tasks, demonstrating its capability to handle nontrivial interaction dynamics across physical domains.
>
> To further support community development, we will release all datasets used in our experiments after publication, enabling future work to benchmark on these more realistic FSI settings.

---

> > ### Comment · Reviewer_yad9 · 2025-11-26
> >
> > Thank you for the detailed response and the additional context regarding the baselines.
> >
> > I have a follow-up question regarding references [2, 6]. It does not appear that these works utilize Transformer architectures to model the specific type of interaction dynamics I was referring to, such as, MPM/SPH datasets where the spatio-temporal evolution is explicitly driven by inter-particle collisions during rollout.
> >
> > Could you clarify if such settings are considered out of the scope of this paper? If so, I would suggest explicitly defining this boundary in the problem setting. Alternatively, if this is a limitation of the current approach, it would be valuable to mention it as an avenue for future work.
> >
> > Additionally, regarding the comparison with MeshGraphNets (MGN): given that MGN is inherently designed for mesh-based simulations, it generally does consider systems with large topological changes or complete deformations (e.g., fluids, sand), despite architectural similarity to GNS. I would appreciate some further clarification on how the paper positions itself regarding these types of materials.
> >
> > Thank you again for the engagement

---

> > > ### Author Response · Authors · 2025-11-26
> > >
> > > Thank you very much for taking the time to review our first-round rebuttal and for providing your follow-up comments. We are glad to continue the discussion.
> > >
> > > Firstly, we sincerely apologize for the misunderstanding in our previous reply. In the first round, we incorrectly assumed that the Lagrangian mesh points used in the works we cited before [2, 6] were conceptually equivalent to the Lagrangian particles you referred to. We now fully understand that your comment was specifically targeting **MPM&SPH material particles**, which carry explicit physical meaning like collision-based interactions, and we appreciate the clarification.
> > >
> > > Regarding this category of particle-based physical systems, we would like to acknowledge that **there indeed exist Transformer-based approaches that directly address MPM/SPH-like particle interactions**. For example, work [8] employs a Transformer structure to model several classic GNS scenarios such as FluidFall, FluidShake, and BoxBath, demonstrating that Transformer architectures are capable of handling explicit particle-interaction-driven rollouts problems. The main reason is that **Transformers can naturally operate on both mesh-point and particle-based inputs**, since these two representations share essentially the same format, each element is characterized by its coordinates along with associated physical quantities. Moreover, the **attention mechanism inherently enables the model to learn interactions between particles**, such as collision effects.
> > >
> > > Regarding our design, we acknowledge that due to the design of interface component, our model can directly handle **some** particle-based scenarios (like box, cube), while **other types** (like sand) may require additional processing. This can also response the question you pointed out between our method and the class of problems addressed by MGN. Recall that we treat the **fluid–solid interface as an independent component**, which implicitly assumes that the solid surface is generally **coherent and possesses sufficient rigidity**. Under this assumption, our method can naturally handle scenarios in GNS that involve rigid or quasi-rigid solids, such as boxes or cubes, since the interface remains well-defined throughout the rollout. However, for materials like **sand**, which have low rigidity and may easily disperse, the interface becomes ill-defined. For example, when water infiltrates sand, the interface between the “solid” and the “fluid” regions is no longer clear, making it difficult to define or maintain an interface component in our current formulation. Consequently, our model cannot be directly applied to such highly deformable or granular materials without additional mechanisms. We will clearly articulate this distinction in the revised manuscript and highlight it as one of the directions for future improvement.
> > >
> > > Thank you for further clarifying you concern and we are glad to think deeply about our design.
> > >
> > > [8] Shao, et al. "Transformer with Implicit Edges for Particle-based Physics Simulation." ECCV2022.

---

> > > > ### Comment · Reviewer_yad9 · 2025-11-26
> > > >
> > > > Thank you for the clarifications. I am convinced and will increase the score. However, I think the clarity and positioning of this work can be improved by adding a discussion section to distinguish this work from neural operator models. Overall, this is an interesting approach

---

> > > > > ### Author Response · Authors · 2025-11-27
> > > > >
> > > > > Thank you for your recognition of our work. Following your suggestions, we will include a discussion between our design and neural operators in the revised version to more clearly position our approach. Thank you once again, sincerely.

---

> ### Author Response · Authors · 2025-11-21
> **Response 3**
>
> > **Question 3**: The rationale behind KNN based grid is not fully clear to me. With KNN based grids, the aggregation region changes based on discretization, how does this enforce discretization convergence.
>
> Firstly, in practice we do **not observe convergence or stability issues** arising from kNN neighborhoods. The grid update operates robustly during training, and similar designs have also been adopted successfully in prior neural-operator works [2]. Choosing kNN instead of radius neighbors can be interpreted from a **sparsity and efficiency perspective**. Instead of aggregating over all points within a radius, we only assign non-zero weights to the **top-k most relevant neighbors** in that region, effectively setting the remaining weights to zero. This is conceptually similar to the sparsity mechanisms used in attention pruning, top-k sparse attention, or mixture-of-experts (MoE) models, where limiting computation to the most informative entries provides both stability and efficiency. Under this view, kNN acts as a controlled sparse operator that preserves the locality structure required by the ALE-inspired smoothing, while avoiding dense neighborhoods that would increase computational cost or introduce numerical instability.
>
> Secondly, the goal of the latent ALE grid is **not to approximate a classical discretization of the PDE operator** (as in GINO or multipole graph kernel networks), but to provide a *learned, geometry-aware latent space* onto which heterogeneous domains (fluid, solid, interface) are unified. In this setting, enforcing strict radius-based coverage is less critical than in standard neural operators, because the latent ALE grid is **not strictly tied to the input discretization**. Its topology remains fixed, and only its geometry is deformed via the ALE-style offset and mesh-velocity updates. This means that the latent grid itself provides a stable discretization backbone, independent of the sampling density of the input points. Moreover, the geometry-aware initialization in $\underline{\text{Section 3.1 of original version}}$ further stabilizes the use of kNN. Since the latent ALE grid is constructed through a **distance–weighted (inverse-distance) aggregation** over the input geometry, the resulting latent grid forms a *density-smoothed* and *regularized* geometric manifold. This smoothing effect ensures that the grid points do not concentrate irregularly even when the input discretization is highly non-uniform. Under such a smoothed geometry, applying kNN produces **stable and well-conditioned neighborhoods** in almost all practical cases, as the local sampling density varies smoothly and the nearest-neighbor structure does not degenerate. This synergy between the geometry-aware offset and the kNN sparsification leads to a consistent and robust neighborhood definition throughout training and inference.
>
> > **Question 4**: The approach inherently handles multi-scale inputs. It would be interesting to see if it also handles AMR, as part of future work.
>
> We truly appreciate the opportunity to discuss future directions, as this is exactly where many of the exciting research ideas emerge. There are many techs simply referred to as AMR. In this context, we interpret it as **Adaptive Mesh Refinement**, a classical technique used in numerical solvers to dynamically increase resolution in regions. If it does not match your intention, we warmly welcome clarification.
>
> We believe our framework is naturally compatible with AMR-like extensions. Since the latent ALE grids are constructed independently of the original discretization and updated through ALE-inspired smoothing, one could dynamically adjust the grid density in regions where deformation, interface motion, or fluid gradients are strongest, without breaking the overall architecture. Moreover, when combined with our multi-scale design, the framework can simultaneously capture information across different resolution levels, allowing adaptive grids of varying densities to contribute coherently to the overall representation. Exploring such adaptive latent grids would be a promising direction for future work. Thank you for highlighting this interesting avenue.
>
> ## Reference
>
> [1] Wu H, et al. "Solving high-dimensional pdes with latent spectral models." ICML 2023.
>
> [2] Tao S, et al. "Unisoma: A Unified Transformer-based Solver for Multi-Solid Systems." ICML 2025.
>
> [3] Hao Z, et al. "GNOT: A general neural operator transformer for operator learning." ICML 2023.
>
> [4] Sanchez, et al. "Learning to simulate complex physics with graph networks." ICML 2020.
>
> [5] Pfaff, et al. "Learning Mesh-Based Simulation with Graph Networks." ICLR 2021.
>
> [6] Yu X, et al. "Point-bert: Pre-training 3d point cloud transformers with masked point modeling."  CVPR 2022.
>
> [7] Rahman, et al. "Pretraining codomain attention neural operators for solving multiphysics pdes." NeurIPS 2024.

---

> ### Comment · Reviewer_yad9 · 2025-11-26
>
> I thank the authors for the clarification regarding the stability and empirical benefits of the kNN implementation.
>
> However, I have a follow-up regarding the theoretical positioning of the method. The paper discusses "Neural Operators" extensively (e.g., the comparison with CoDA-NO) and critiques existing operators. In the context of Operator Learning, a defining characteristic is discretization invariance ie. the ability to approximate a continuous operator independent of the mesh resolution.
>
> As noted in the operator literature (e.g., the GINO paper [Li et al., Kovachki et al.]), kNN-based graphs are inherently discretization-dependent. As the sampling density increases, the physical extent of a kNN neighborhood shrinks, meaning the receptive field effectively changes with resolution. This contrasts with radius-based graphs, which are necessary to approximate continuous integral operators.
>
> Could the authors clarify if they claim discretization invariance (in the strict Operator Learning sense) for this architecture? If so, how is the mesh-dependency of kNN reconciled with the integral operator definition? If not, it might be beneficial to refine the terminology to distinguish this approach from resolution-independent Neural Operators.
>
> It would additionally be helpful to discuss KNN + distance weighted aggregation in comparison to radius based integral transform. Are they mathematically equivalent?
>
> Neural Operator: Learning Maps Between Function Spaces With Applications to PDEs, Nikola Kovachki

---

> > ### Author Response · Authors · 2025-11-26
> >
> > We thank the reviewer for raising the important point about discretization invariance of Neural Operators.
> >
> > We first briefly recall the pipeline of our algorithm. We begin by initializing a **regular grid**, and then perform a **distance-weighted aggregation** over the mesh points of different domains to obtain **geometry-aware offsets**, which are used to initialize the **latent ALE grid**. Next, we construct **kNN neighborhoods** on the latent ALE grid to determine the local neighbor sets for each grid node. Based on the initialized latent ALE grid, we project physical quantities of original mesh points onto the latent ALE grid and then apply **PCM** to learn the coupling between different components, where the kNN-based neighborhoods are primarily used to model the **motion of the latent ALE grid** itself. Finally, we perform **inverse interpolation** to recover the next state of the mesh points in the original physical space.
> >
> > According to the strict definition of Neural Operators, our current implementation, which is based on kNN rather than radius-based neighborhoods, is indeed **not resolution-independent**. However, we would like to further clarify the following points:
> >
> > - The kNN neighborhood is used purely as a latent-space topological structure to support latent ALE mesh motion. Since the latent ALE grids are constructed with a geometry-aware offset, the effective neighborhood remains sufficiently stable for our purposes, even though its physical extent varies with sampling density.
> > - Our main focus is on building a unified, geometry-aware representation for heterogeneous fluid–solid domains and learning their bidirectional coupling. Our use of kNN based on a fixed number of neighbors (rather than a fixed physical radius) is primarily motivated by computational efficiency, and a radius-based construction would also be a feasible choice within our framework.
> >
> > Given the flexibility of neighborhood choice and the formal definition of Neural Operators, we therefore position our model as an **operator-inspired, cross-domain FSI framework**, rather than a **fully discretization-invariant Neural Operator**. We will explicitly clarify this distinction in the revised manuscript.
> >
> > Regarding whether **distance-weighted aggregation + kNN** is *mathematically equivalent* to a **radius-based integral operator**, we apologize that it is hard for us to provide a rigorous mathematical proof within the scope of this rebuttal. However, we can offer an intuitive analysis and discussion:
> >
> > The distance-weighted aggregation acts like an **kernel-based aggregation**, assigning larger weights to nearby grids and smaller weights to distant ones. The kNN on the latent ALE grid introduces an explicit **local neighborhood constraint** that governs diffusion and propagation. Functionally, the combination of these two components plays a role similar to radius-based integral operators: they integrate information through (1) an aggregation kernel, and (2) a locality constraint enforced by neighborhood structure. If we truncate the distance-based kernel, each latent grid node effectively aggregates information only from points within its nearest-distance region. In this setting, an appropriately chosen $k$ can be regarded as performing an integral over the points within a small neighborhood in the original space. This combination can be viewed as an **approximation** to a radius-based integral operator. In practice, we did not apply an explicit distance truncation nor carefully tune the value of $k$, yet the model still achieves strong empirical performance. We believe that developing a deeper theoretical understanding of this behavior is valuable, and we will further investigate and strengthen the theoretical aspects in future work. Thanks again for your valuable and constructive suggenstions.

---

### Author Response · Authors · 2025-11-21
**Revision**

We sincerely thank all reviewers, as well as the ACs, SACs, and PCs, for the time and effort spent evaluating our work and for the valuable comments and suggestions provided. Based on this feedback, we have revised the paper to further strengthen the clarity and quality of the work. Our main modifications are as follows:

1. Advised by reviewer yad9 and yXgU, we removed redundant descriptions of detailed operation and added clearer intuition and insight regarding our use of the attention mechanism (Section 3.3).
2. Advised by reviewer yXgU, we further clarified the model inputs (Section 3), added descriptive subcaptions to Figure 2 and Table 3 for improved readability, and moved the *Limitations*  from the appendix to the main paper (Section 5).
3. Advised by reviewer k2K6, we added a new subsection that visualizes and analyzes the attention patterns, providing additional interpretability insights (Section K).

We use blue text to highlight the modification. We hope these revisions address the reviewers’ concerns. We also look forward to further in-depth and detailed discussions, which will undoubtedly help us continue improving the quality of our work.

---

### Meta-Review · Program_Chairs · 2026-01-06

**Summary:**

authors propose Fisale, a data-driven surrogate model for two-way Fluid-Structure Interaction (FSI). The framework explicitly models fluid, solid, and interface components by projecting them onto multi-scale latent Arbitrary Lagrangian-Eulerian (ALE) representations. It then performs iterative updates via a solver-inspired Partitioned Coupling Module (PCM). While reviewers initially raised concerns regarding theoretical soundness and discretization invariance, the authors successfully demonstrated that their structured architectural choices outperform standard, monolithic AI methods.

**Reviewer Concerns:**

Concern 1: Reviewers yad9 and yXgU found Section 3 too dense and felt it described standard ML operations without providing physical intuition.



Resolution: Addressed. The authors revised the manuscript to remove redundant operational details (like basic attention math) and added clear physical insights. For example, they explicitly linked the Partitioned Coupling Module (PCM) to classical partitioned FSI solvers and explained why attention is suitable for capturing both short-range and long-range pressure distributions.



Concern 2: Reviewer yad9 pointed out that using $k$NN-based neighborhoods makes the model discretization-dependent, which contradicts the strict definition of a "Neural Operator" (which requires resolution independence).



Resolution: Addressed via Refinement. The authors conceded that the model is not strictly resolution-independent in the "Operator Learning" sense. They updated the paper's positioning to describe Fisale as an "operator-inspired framework" rather than a fully discretization-invariant operator. This nuance satisfied the reviewer’s theoretical concerns.



Concern 3: Reviewer yad9 asked how the model compares to GNS for multi-material systems like sand or water.



Resolution: Clarified Limitation. The authors clarified that Fisale assumes a coherent interface. While it excels at rigid/deformable solids (boxes, wings), it is not currently designed for "ill-defined" interfaces like water infiltrating sand. They added this as an explicit boundary in the "Limitations" section.



Concern 4: Reviewer yXgU suggested improvements might just come from higher parameter counts ("overfitting") rather than the ALE or PCM designs.



Resolution: Addressed via Ablation. The authors pointed to Table 16, showing a ~10% performance drop when the PCM was replaced by a simpler module, even when keeping parameters constant. They also highlighted that Fisale uses fewer parameters than several baselines while achieving better results.



Concern 5: Reviewers yXgU and k2K6 noted the model is trained separately for each task (Structure Oscillation, Venous Valve, etc.) and questioned its ability to learn "physical laws" vs. just fitting data.



Resolution: Partially Addressed. The authors admitted that task-level generalization is a broad challenge for the field. However, they provided Out-of-Distribution (OOD) results (Appendix H) showing the model generalizes to unseen physical parameters (like different Reynolds numbers) within a task, proving it isn't just memorizing specific samples.



Concern 6: Reviewer wVhk questioned why Linear Attention was used when Standard Attention (or FlashAttention) might be more accurate for the scale of these problems.



Resolution: Addressed via Benchmark. The authors provided a head-to-head comparison in the rebuttal. They showed that Standard Attention provided no significant accuracy boost but was slower and more memory-intensive. This confirmed that Linear Attention was the better choice for scaling to larger FSI problems.

**Reviewer Scores:**

The authors were highly effective in their rebuttal. By providing new empirical data, head-to-head attention benchmarks, and theoretical concessions, they moved the needle significantly:



• Reviewer yad9: 4 $\rightarrow$ 6



• Reviewer wVhk: 6 $\rightarrow$ 8



• Reviewer k2K6: Maintained 8



• Reviewer yXgU: It may be upgraded 2 to 4.

---

### Decision · Program_Chairs · 2026-01-26

Accept (Poster)